# Radargrammetric DSM Generation by Semi-Global Matching and Evaluation of Penalty Functions

Jinghui Wang [1,2], Ke Gong [2], Timo Balz [1,*], Norbert Haala [2], Uwe Soergel [2], Lu Zhang [1] and Mingsheng Liao [1]

1   State Key Laboratory of Information Engineering in Surveying, Mapping and Remote Sensing (LIESMARS), Wuhan University, Wuhan 430072, China; wjh@whu.edu.cn (J.W.); luzhang@whu.edu.cn (L.Z.); liao@whu.edu.cn (M.L.)
2   Institute for Photogrammetry, University of Stuttgart, Geschwister-Scholl-Strasse 24, 70174 Stuttgart, Germany; ke.gong@ifp.uni-stuttgart.de (K.G.); norbert.haala@ifp.uni-stuttgart.de (N.H.); uwe.soergel@ifp.uni-stuttgart.de (U.S.)
*   Correspondence: balz@whu.edu.cn

**Abstract:** Radargrammetry is a useful approach to generate Digital Surface Models (DSMs) and an alternative to InSAR techniques that are subject to temporal or atmospheric decorrelation. Stereo image matching in radargrammetry refers to the process of determining homologous points in two images. The performance of image matching influences the final quality of DSM used for spatial-temporal analysis of landscapes and terrain. In SAR image matching, local matching methods are commonly used but usually produce sparse and inaccurate homologous points adding ambiguity to final products; global or semi-global matching methods are seldom applied even though more accurate and dense homologous points can be yielded. To fill this gap, we propose a hierarchical semi-global matching (SGM) pipeline to reconstruct DSMs in forested and mountainous regions using stereo TerraSAR-X images. In addition, three penalty functions were implemented in the pipeline and evaluated for effectiveness. To make accuracy and efficiency comparisons between our SGM dense matching method and the local matching method, the normalized cross-correlation (NCC) local matching method was also applied to generate DSMs using the same test data. The accuracy of radargrammetric DSMs was validated against an airborne photogrammetric reference DSM and compared with the accuracy of NASA's 30 m SRTM DEM. The results show the SGM pipeline produces DSMs with height accuracy and computing efficiency that exceeds the SRTM DEM and NCC-derived DSMs. The penalty function adopting the Canny edge detector yields a higher vertical precision than the other two evaluated penalty functions. SGM is a powerful and efficient tool to produce high-quality DSMs using stereo Spaceborne SAR images.

**Keywords:** radargrammetry; SAR; semi-global matching; penalty function

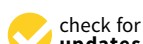

## 1. Introduction

A Digital Surface Model (DSM) is a fundamental dataset in remote sensing applications. Synthetic Aperture Radar (SAR) images can be used to construct Digital Surface Models. Four methods can be used to extract height from SAR images: clinometry, interferometry, radargrammetry, and polarimetry [1]. InSAR (Interferometric Synthetic Aperture Radar) and stereo radargrammetry however, are the most widely used techniques among them. For instance, SRTM DEM [2] and TanDEM-X DEM [3] are the most representative InSAR products, while StereoSAR DSMs are generated from SAR images with different incidence angles.

Depending on phase information of SAR imagery, the InSAR technique is more prone to decorrelation related to temporal changes in objects, large spatial baselines, or turbulent atmospheric conditions, hampering the achievable DSM quality. Radargrammetry however, is an alternative to InSAR processing for DSM retrieval as it makes use of amplitude information and thus is less susceptible to the decorrelation effect. For a given pair of images,

usually either an InSAR or radargrammetry technique can be applied: the former requires small changes in illumination angle to acquire repeat-pass images, whereas the latter requires large baselines to yield stable intersection of line-of-sight rays. Radargrammetry is analogous to photogrammetry where disparities of correspondences on stereo images are computed by image matching and used to derive 3D point clouds. Therefore, the accuracy of image matching directly determines the quality of final DSMs. Much research has been devoted to stereo image matching.

According to Scharstein and Szeliski [4], stereo matching methods can be divided into two categories: local matching and global matching algorithms. In addition, stereo matching algorithms are generally composed of four steps: matching cost calculation, cost aggregation, disparity computation, and disparity refinement. Based on this taxonomy and distinctions of the algorithmic building blocks, the performance of a variety of stereo algorithms can be effectively evaluated and improved.

Local matching algorithms, namely, window-based algorithms, compute disparity of correspondences utilizing only the intensity values within a finite window centered by the point to match. Usually, local algorithms consist of two steps: matching cost computation and cost aggregation. Matching cost is computed from area-based similarity measures. Common similarity measures include normalized cross-correlation (NCC), sum of absolute differences (SAD), sum of squared differences (SSD), as well as census transformation [5], and mutual information (MI) [6]. Cost aggregation is realized by operation on the intensity values within a support window such as box-filters or weighted sums. Disparity is determined simply with the minimum matching cost by the "winner-take-all" (WTA) strategy. Local algorithms have high computational efficiency but with several disadvantages. The use of a support window implicitly assumes that the disparities between the pixels within the window are consistent. This violates real world conditions as discontinuities are ubiquitous; local algorithms yield sparse and inaccurate correspondences and derive ambiguous results. Moreover, the obtained disparity for each pixel is locally optimized; no matter if a fixed or an adaptive window is selected. The size of the window also influences matching performance.

Global matching algorithms, however, use all the information from all the pixels in the image to estimate disparities. Typically, cost aggregation is skipped and disparity computation is the key step. Global algorithms are formulated as an energy function, combining a data term and a smoothness term. The data term is composed of matching costs, and the smoothness term is an explicit expression of shape priors, which can be specifically designed. By introducing the smoothness term, global algorithms can implement regularization constraints to adjacent pixel disparities to preserve discontinuities. Therefore, global algorithms behave robustly even in occlusive, discontinuous, or textureless regions. Disparities are obtained when the global energy is at the minimum. This, however, is an NP-hard problem [7], thus optimization methods such as belief propagation (BP) [8] and graph cut (GC) [7,9] are used to approximate the minimum solution of the global energy. The disparity quality of global algorithms is precise but at a high computational and memory cost.

The semi-global matching (SGM) algorithm proposed by Hirschmüller [10–12] achieves a compromise between accuracy and computational efficiency. SGM approximates the 2D energy optimization problem by aggregating cost along several 1D scanlines, greatly reducing the processing time. Pixel-wise disparity is determined by adding up matching costs from all directions and selecting the disparity where the cost is minimal. Beyond its accuracy and fast matching speed, the ability to retrieve homogeneous surfaces as well as preserve sharp height discontinuities makes SGM one of the most effective matching algorithms. Modifications to the original SGM are an active area of research and there are a number of modified versions of SGM like tSGM [13], weighted SGM [14], iterative SGM [15], SGM forest [16], object-based SGM [17], and so forth.

In terms of radargrammetric DSM generation, local matching has been the dominant matching approach since the 1980s [18–22]. No work has been published to investigate the

feasibility and effectiveness of performing pixel-wise dense matching methods for more than two decades. Until 2014, a hierarchical semi-global matching was used to retrieve DSM from TerraSAR-X stereo imagery, demonstrating that the epipolar geometry concept widely used in optical stereo methods could also be applicable to satellite SAR stereo pairs [23]. The authors also applied the same approach to TerraSAR-X staring Spotlight triplets [24] for 3D mapping. They further improved the epipolar rectification method in [25] where the SAR image pairs are projected onto a coarse a priori DEM (e.g., SRTM, ASTER) instead of a plane. However, the work in [23–25] focused on the study of the space-borne SAR epipolar model, and we can only know some major parameters and that a constant penalty function was adopted from [23], no parameters or details of the SGM algorithm were presented in [24] or [25].

In 2016, Di Rita et al. developed a radargrammetric DSM workflow termed the DATE workflow [26–28] that exploits the OpenCV library Semi-Global Block Matching (SGBM) algorithm [29,30]. The workflow also implements a coarse-to-fine matching method, which computes a disparity map on each pyramid level. The DATE workflow does not transfer disparity from the higher level to a lower level to reduce the disparity search range for a lower-level matching; instead, it converts the disparities to height correction values at every level. The final DSM is obtained by iteratively adding height corrections to a coarse a-priori DSM at each pyramid level. The core of the DATE workflow relies on the a priori DSM hierarchically refined to generate a final DSM. In addition, the SGBM algorithm in OpenCV sets a fixed disparity search range for all pixels, which could lead to more outliers and voids in the output disparity map. Furthermore, users must set and adjust the minimum disparity value, the disparity range, and the matched block size [31], all of which influence the final DSM accuracy. In 2018, Bagheri et al. proposed a semi-global matching framework to investigate the potential of 3D reconstruction from SAR-optical image pairs and feasibility of generating point clouds with a median accuracy [32], but the framework is designed for extracting sparse homologous points from SAR-optical stereo pair and not for DSM reconstruction.

In this study, we propose a hierarchical SGM dense matching pipeline to generate high-quality radargrammetric DSMs in densely vegetated and mountainous areas. The feasibility and effectiveness of the semi-global dense matching algorithm is investigated using Stripmap and Spotlight mode TerraSAR-X stereo data pairs covering Mount Song in central China. Furthermore, we investigated the influence of three penalty functions on the vertical accuracy of final DSMs. The pipeline was built in a user-friendly manner where only two penalty parameter values need to be set. Without any post-processing, the disparity map can be directly used for generating final DSMs. A high-resolution airborne photogrammetric DSM was used to validate the radargrammetric DSMs. NASA's 30 m resolution SRTM DEM and DSMs extracted by hierarchical NCC matching approach were also included to compare the vertical accuracy. The results demonstrate that, our SGM pipeline not only produces DSMs with higher vertical accuracy than the SRTM DEM and NCC-derived DSMs, but is also more efficient than the NCC local matching method. In addition, a penalty function exploiting the Canny edge detector delivers higher vertical accuracy than the constant penalty function, or the gray gradient penalty function in stereo SAR semi-global matching. Semi-global matching is a powerful alternative tool to local matching algorithms with higher accuracy and efficiency for radargrammetric DSM generation in complex mountainous areas.

## 2. Materials and Methods

We implemented a hierarchical semi-global matching method—a tSGM-like method, for matching stereo SAR images. After preprocessing the SAR images, epipolar images are generated from stereo SAR amplitude images, and Gaussian pyramid images are derived from the epipolar pair. Consequently, SGM is executed on each pyramid level to produce a disparity map. The disparity map of a higher level is transferred to a lower level, dynamically shortening the disparity search range for every pixel at the lower

level. Disparities of the lowest pyramid level form the final desired disparity map. Image coordinates of homologous points are calculated from the output disparities. In turn, these homologous points are intersected based on the range-Doppler model to derive 3D geographic coordinates, so the coordinates are used to yield a final gridded DSM by a triangulation interpolation. The formulation of the penalty function as expressed in Equation (1) influences the matching performance (discussed in Section 2.4). Thus, we encoded three different formulations of the penalty parameter $P_2$ in the hierarchical SGM pipeline and obtain corresponding disparity maps to assess the impact of different penalty functions on the final DSM accuracy. In addition, we implemented a hierarchical NCC local matching algorithm for comparison. In this section, we interpret principles and fundamental procedures of these matching algorithms.

### 2.1. Mount Song Test Site and Datasets

The test site is situated at Mount Song in the northwest of Henan province in China, as shown in Figure 1. Mount Song is one of the prominent cradles of Chinese civilization and is well-known as the central mountain of the Five Great Mountains of China. With height ranging from 150 m to 1512 m above sea level, Mount Song possesses a complex terrain. Mount Song, with 3.6 billion years of geological history, is composed of metamorphic and sedimentary rocks from five geologic periods including the Archean, Proterozoic, Paleozoic, Mesozoic, and Cenozoic epochs. The mountain massif stretches from east to west and constitutes steep slopes and precipitous cliffs.

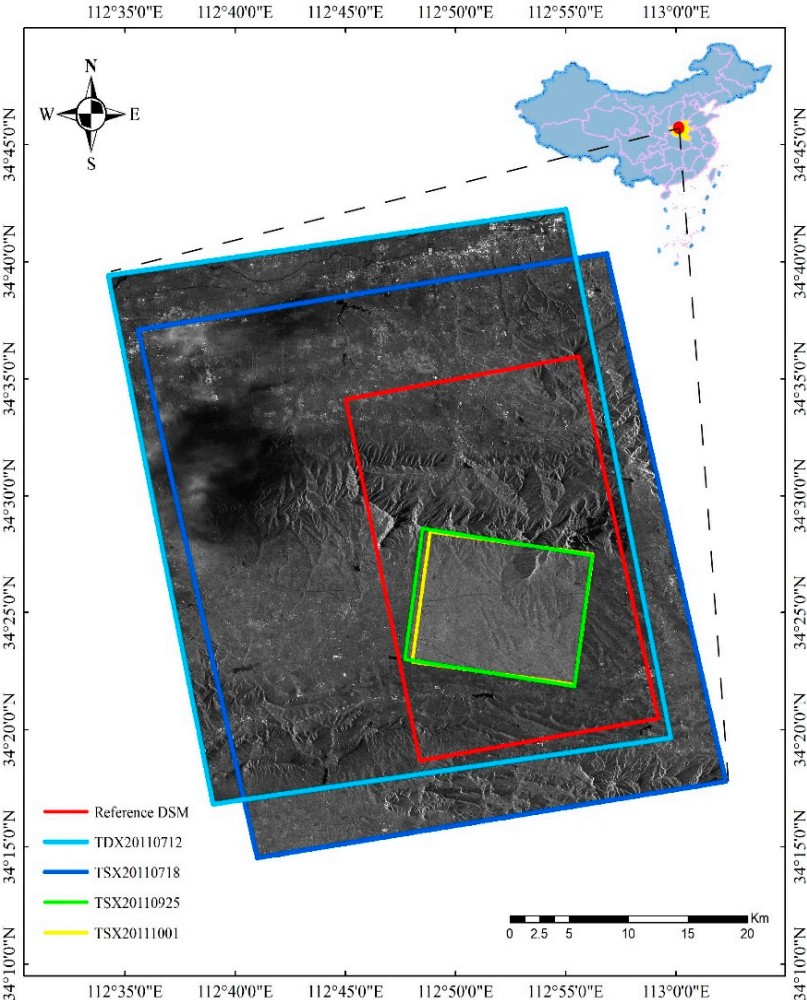

**Figure 1.** The Mount Song test area with coverage of photogrammetric reference DSM and TerraSAR-X/TanDEM-X stereo images.

Our study area is covered with vegetation, thus the DEM generation by the InSAR approach is heavily affected by decorrelation, and therefore, radargrammetry is a suitable way to map this area. We selected two pairs of stereo TerraSAR-X/TanDEM-X images to generate DSMs for the Mount Song area. One was a Stripmap (SM) stereo pair (SM_0712, SM_0718) collected from an ascending orbit; the other was a Spotlight (SL) stereo pair (SL_0925, SL_1001) collected from a descending orbit. The image acquisition parameters are listed in Table 1. The coverage of the stereo images is shown in Figure 1.

**Table 1.** Acquisition parameters of TerraSAR-X/TanDEM-X stereo images.

| ID | Satellite | Acquisition Time | Acquisition Mode | Orbit Direction | Incidence Angle (°) | Resolution rg/az [1] (m) |
|---|---|---|---|---|---|---|
| SM_0712 | TDX-1 | 12 July 2011 | SM | Ascending | 44.5 | 1.8/3.3 |
| SM_0718 | TSX-1 | 18 July 2011 | SM | Ascending | 28.9 | 1.2/3.3 |
| SL_0925 | TSX-1 | 25 September 2011 | SL | Descending | 32.2 | 1.2/1.6 |
| SL_1001 | TSX-1 | 1 October 2011 | SL | Descending | 47.1 | 1.2/1.6 |

[1] rg is slant range and az is azimuth.

As shown in Figure 1, one Stripmap image is shielded by heavy rain, which affects the stereo matching. The influence is seen in the final radargrammetric DSMs generated from this Stripmap pair. An airborne photogrammetric DSM at 1 m resolution with a height precision of 1 m was used as the reference data for evaluating the quality of our radargrammetric DSMs. Although it was acquired in 2009, two years earlier than the stereo SAR image acquisition time, the study area remained relatively stable and demonstrated no big changes so that the reference DSM is sufficient to validate our radargrammetric method.

NASA's 30 m spatial resolution SRTM DEM [2] acquired with InSAR approach was also used for the height accuracy comparisons with the DSMs generated with our radargrammetric methods. The SRTM DEM and the radargrammetric DSMs are comparable because they were all generated from X-band SAR images, thus the extracted heights all refer to the upper surface of objects on earth. The SRTM DEM was also used in the production of the epipolar images for stereo matching, as discussed in the next subsection.

### 2.2. Epipolar Rectification

Epipolar images are the input data required for the SGM pipeline. After epipolar rectification, correspondences are aligned on epipolar lines so that the 2D image matching problem is simplified to 1D matching along epipolar lines. We applied multi-looking and Lee filtering to attenuate the inherent speckle and noise in SAR amplitude images, then adopted the projection trajectory method assisted by external DEM, proposed by Perko et al. [25], to generate rectified epipolar images in object space. For the Stripmap mode images with the range resolution of 1.8 m and the azimuth resolution of 3.3 m, we performed $3 \times 3$ multi-looking on the amplitude images. For the Spotlight mode images with the range resolution of 1.2 m and the azimuth resolution of 1.6 m, we performed $5 \times 5$ multi-looking on the amplitude images. Therefore, the epipolar images of both stereo pairs were generated with a spatial resolution of 10 m, and DSMs were derived from both stereo pairs with the same resolution of 10 m. If amplitude images at full resolution are used to generate epipolar images, the matching process will introduce many mismatched disparities or voids due to the speckles and noise maintained in SAR images. The a priori SRTM DEM helps eliminate most of the SAR geometric distortions in range direction including layover and foreshortening effects. When such a priori DEMs are not available, we can generate them with commonly-used radargrammetric methods such as NCC matching, or simply by applying an identical mean height value as proposed by [23]. Discussion of epipolar geometry, however, is not the focus of this study and readers can refer to [23,25,33] for further details. We show the stereo anaglyphs of epipolar images generated from our TerraSAR-X stereo pairs in Figure 2.

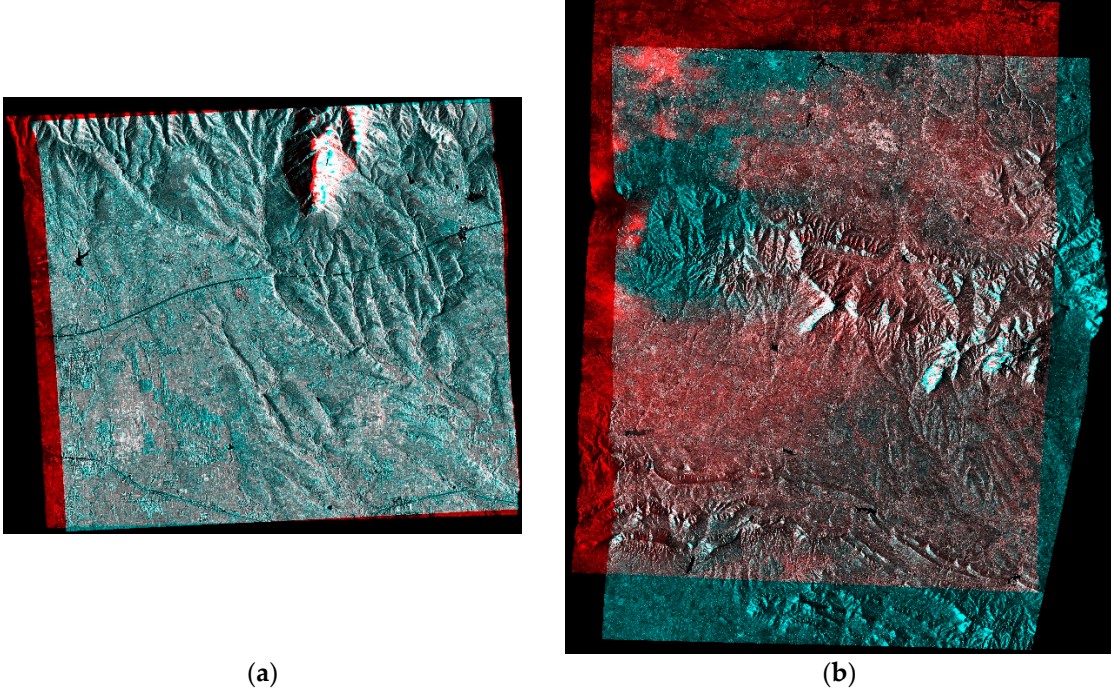

(**a**)

(**b**)

**Figure 2.** (**a**) Stereo anaglyph of Spotlight epipolar images; (**b**) Stereo anaglyph of Stripmap epipolar images.

In Figure 2, the size of Spotlight epipolar images in (a) is 1650 × 1689 pixels, and the size of Stripmap epipolar images in (b) is 5902 × 5261 pixels. The blue and red colors represent the disparity between the left and right epipolar images. Only one color showing in the overlapping areas indicates that there is no disparity between the homologous points, and most of the disparities were eliminated in each epipolar pair. The white regions refer to areas that lack matching texture. This will be shown in Section 3.3.2. To further illustrate the geometric consistency of the epipolar images, a zoomed region was selected from each epipolar pair and shown in Figure 3.

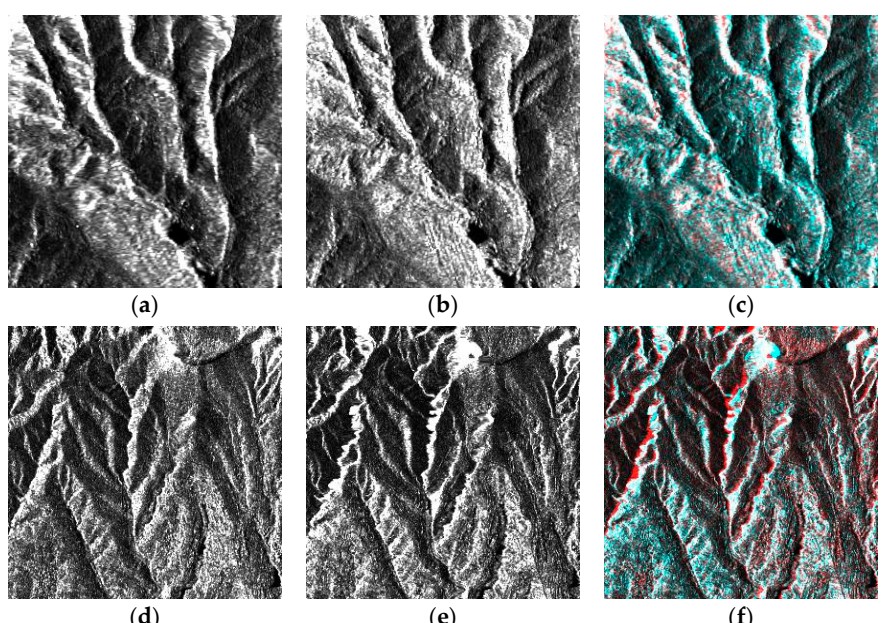

(**a**)　　　　　　　(**b**)　　　　　　　(**c**)

(**d**)　　　　　　　(**e**)　　　　　　　(**f**)

**Figure 3.** Zoomed region in (**a**) left and (**b**) right Spotlight epipolar image and (**c**) the stereo anaglyph. Zoomed region in (**d**) left and (**e**) right Stripmap epipolar image and (**f**) the stereo anaglyph.

Figure 3 shows most of the homologous points match in the stereo anaglyphs, appearing as the same color. Only slight residual disparities remain along the epipolar direction shown in the red or blue color. In this case, the disparity search range shrinks. Thus, stereo matching is easier and faster.

*2.3. Hierarchical Semi-Global Matching*

The semi-global matching algorithm models the correspondence problem as an energy function which combines a data term and a smoothness term, in the same way as global matching algorithms. Semi-global matching minimizes a global cost function to extract pixel-wise disparity based on the following form:

$$E(D) = \sum_p (C(p,\ D(p))) + \sum_{q \in N_p} P_1 T[|D(p) - D(q)| = 1] \\ + \sum_{q \in N_p} P_2 T[|D(p) - D(q)| > 1] \tag{1}$$

where $D$ represents the disparity image with the same size of epipolar images, in which the value of each pixel is the disparity of corresponding pixel in the base image. $E$ is the global energy to be minimized by finding optimal disparities. To the right of the equal sign, the first term is the data term summing up matching cost of every pixel where $p$ represent the pixel on base image and $C$ is the pixel-wise photo consistency measure. The second term is one part of smoothness term regularizing small disparity changes where pixel $q$ is in the vicinity of pixel $p$. Operator $T$ produces a value of one if the subsequent condition is true and otherwise produces a zero value. The third term is the other part of smoothness term controlling large disparities to maintain height discontinuities. The smoothness term forces a small constant penalty $P_1$ when the disparity difference of pixel $p$ and its neighbor pixel $q$ equals to one, and a large constant penalty $P_2$ when the disparity difference of pixel $p$ and its neighbor pixel $q$ exceeds one.

2.3.1. Matching Cost Calculation

The first step of SGM is calculating matching cost for every pixel on the rectified base image. Given a disparity $d$, the photo consistency is computed for pixel $p$ on the left epipolar image and pixel $p + d$ on the right epipolar image. Census transformation [5] was chosen as the photo consistency measure because it delivers the most robust matching cost results among a variety of parametric and non-parametric matching cost methods, and the mutual information cost method [34].

We adopted a $9 \times 7$ window to perform the census transform for every pixel. It compares the amplitude value of each pixel within the window with the center pixel to produce a binary string. By performing exclusive-OR operation between the binary strings of suspected correspondences, the matching cost was calculated by Hamming distance. For each pixel, the matching cost is calculated for every disparity in its disparity range and accumulated in the cost aggregated step.

2.3.2. Cost Aggregation

SGM aggregates matching costs following the concept of Dynamic Programming. It chooses multiple directions on image to perform the aggregation and finally adds up matching costs of all directions instead of optimizing along only one image row. The original SGM calculates the aggregated cost $L$ for pixel $p$ given disparity $d$ along arbitrary direction $r_i$ by Equation (2):

$$L_{r_i}(p,d) = C(p,d) + \min \left[ L_{r_i}(p - r_i, d),\ L_{r_i}(p - r_i, d - 1) + P_1, \\ L_{r_i}(p - r_i, d + 1) + P_1,\ \min_j L_{r_i}(p - r_i, j) + P_2 \right] - \min_k L_{r_i}(p - r_i, k) \tag{2}$$

where $p - r_i$ is the previous pixel of pixel $p$ along the path.

As we implemented a hierarchical SGM method, we transferred the disparity from a higher pyramid level to a lower pyramid level in order to make an adaptive adjustment of disparity search range for every pixel, which is no longer an identical disparity search range like [–32, 32] for all pixels. Specifically, the disparity map of a higher level is upscaled by two to match the size of the next level, so the disparity value of pixel $p$ is multiplied by two to serve as the initial disparity value $d(p)$ for the same pixel on a lower level. The complete disparity search range for pixel $p$ on the lower level is defined as $[d(p) - 4, \ d(p) + 4]$. Thus, a pixel-wise disparity range is determined for each pixel on the lower level, and the radius of four guarantees that the range is not too large to increase ambiguities or too narrow to miss the correct disparity value. This is a coarse-to-fine matching strategy that decreases outliers by narrowing the disparity search range as well as prompts matching efficiency. Consequently, the aggregated cost is modulated according to Equation (3):

$$
\begin{aligned}
&\text{if } d > d_{max}(p - r_i) : \\
&\qquad L_{r_i}(p,d) = C(p,d) + L_{r_i}(p - r_i, d_{max}(p - r_i)) + P_2 - \min_k L_{r_i}(p - r_i, k) \\
&\text{if } d < d_{min}(p - r_i) : \\
&\qquad L_{r_i}(p,d) = C(p,d) + L_{r_i}(p - r_i, d_{min}(p - r_i)) + P_2 - \min_k L_{r_i}(p - r_i, k) \\
&\text{else:} \\
&\qquad L_{r_i}(p,d) = C(p,d) + \min[L_{r_i}(p - r_i, d), \ L_{r_i}(p - r_i, d - 1) + P_1, \\
&\qquad L_{r_i}(p - r_i, d + 1) + P_1, \ \min_j L_{r_i}(p - r_i, j) + P_2] - \min_k L_{r_i}(p - r_i, k)
\end{aligned}
\tag{3}
$$

The aggregated cost is added up from all directions to obtain the final cost for pixel $p$ at a given disparity $d$:

$$
S(p,d) = \sum_i L_{r_i}(p,d) \tag{4}
$$

### 2.3.3. Disparity Computation and Refinement

To determine the final disparity within the disparity range for pixel $p$, we select the disparity with the minimum aggregated cost in accordance with the "Winner Takes All" approach.

$$
D(p) = arg\min_d S(p,d) \tag{5}
$$

The sub-pixel level disparity is estimated by fitting a quadratic curve through the determined pixel-level disparity and the two disparities corresponding to the aggregated costs on either side of the minimum cost. However, these sub-pixel level disparities contain outliers and mismatched points due to occlusions and weak texture. Left-right consistency check, removal of peaks, and median filtering are conducted according to Hirschmüller [12] to remove unreliable disparities. At this stage, the hierarchical semi-global matching is completed and the final disparity map can be exported. The disparity map can be directly used for spatial forward intersection in the next step without any post-processing.

### 2.4. Penalty Function Schemes for $P_2$

We encoded three penalty functions for $P_2$ in our hierarchical SGM pipeline for evaluation. The three SGM algorithms with each $P_2$ formulation are abbreviated with **SGM_const**, **SGM_gray** and **SGM_canny** hereinafter. The penalty functions to be evaluated are: constant penalty, gray gradient penalty, and Canny edge penalty. The formulations of the three penalty functions are as follows:

(1)    $P_2$ is a constant value determined by experience:

$$
P_2(p) = P_2^0 = const. \tag{6}
$$

(2)   $P_2$ is inversely proportional to the absolute gray/amplitude gradient of the current processed pixel $p$ and its proceeding pixel $p - r$ along the path. The original SGM adopts this penalty formulation for $P_2$:

$$P_2(p) = \max\left( \frac{P_2^0}{|I(p) - I(p - r)|}, \ P_1 \right) \tag{7}$$

(3)   $P_2$ is calculated by the Canny algorithm which is an advanced edge detection algorithm. This penalty function is applied in the tSGM method [13]:

$$P_2(p) = \begin{cases} P_{21}, & C(p) = 1 \\ P_{22}, & C(p) = 0 \end{cases} \tag{8}$$

$C(p)$ is the binary value of the Canny edge detector. $P_{21}$ and $P_{22}$ are empirically determined constant values. When an edge is detected, low penalty $P_2 = P_{21}$ is imposed. Otherwise, when no edge is detected, larger penalty $P_2 = P_{22}$ is applied. In practice, we found that making $P_{21}$ equivalent to $P_1$ could give reliable results for the census matching cost.

To find the optimal parameter values for each penalty function, we used a "coarse-to-fine" strategy to adjust the parameter values. This strategy guarantees that the step size is not too large to miss the optimal value and not too small to bring heavy computation. First, we set the tuning range for $P_1$ as [10, 200] with a step size of 20, and the tuning range for $P_2$ as [50, 1500] with a step size of 100. Notice that for every $(P_1, P_2)$ group, $P_2$ should be larger than $P_1$. In practice, we found that different combinations of $P_1$ and $P_2$ make a big difference on the completeness of the disparity map, i.e., the percent of effective disparity values, while the effective disparity values almost remain the same using different $(P_1, P_2)$ combinations. Therefore, the optimal values of $(P_1, P_2)$ are selected based on the disparity map with the highest completeness. Second, a smaller step size is used to further adjust the optimal $(P_1, P_2)$ values obtained in the first step. The new step size is 10 for $P_1$ and 50 for $P_2$. The final optimal values are determined with the highest disparity completeness.

### 2.5. Hierarchical NCC Matching

For comparison, we also implemented a hierarchical normalized cross-correlation (NCC) matching algorithm for the epipolar pyramids to generate the DSMs for the Song Mountain study area. NCC is still the most common SAR imagery matching method, no matter for DSM reconstruction [22,35,36] or for surface motion estimation by pixel offset tracking technique [37–39]. The hierarchical NCC matching strategy is widely used in radargrammetry. Commercial software products such as PCI-Geometica [40], SISAR [20], and SARscape [41], all implement a hierarchical NCC matching method in their radargrammetric modules.

NCC matching is a typical local algorithm in which correspondences are searched based on the similarity of two match windows. The NCC coefficient $\gamma$ of two window templates is formulated in Equation (9):

$$\gamma(p, q) = \frac{\left| \sum_{(i,j) \in W} (m_{i,j} - \mu_m)(s_{i,j} - \mu_s) \right|}{\sqrt{\sum_{(i,j) \in W} (m_{i,j} - \mu_m)^2} \sqrt{\sum_{(i,j) \in W} (s_{i,j} - \mu_s)^2}} \tag{9}$$

where $p$ is the pixel on the base image for which the NCC coefficient is calculated, $q$ is the pixel with disparity shift $d$ along epipolar direction on the matching image: $q = p + d$; $W$ represents the matching window on the base image and the matching image; $m_{i,j}$ and $s_{i,j}$ are the amplitude values at $(i, j)$ in the matching window of the base and matching image respectively; $\mu_m$ and $\mu_s$ are the mean values of base and match window. For each pixel in the search window on the match image, a NCC coefficient is calculated. Abiding by

the "Winner Takes All" (WTA) rule, the correspondences are determined with the largest NCC coefficient which exceeds a preset threshold. For comparison with SGM results, no post-processing is applied to the NCC disparity map.

In our experiment, we generated a Gaussian pyramid of epipolar images with five levels. Thus, stereo matching was performed on each level and disparities were transmitted from a higher level to a lower level. The main parameters of each stereo matching algorithm are summarized in Table 2.

**Table 2.** Main parameters used in applied matching algorithms.

|  | **NCC** | **SGM_Const** | **SGM_Gray** | **SGM_Canny** |
|---|---|---|---|---|
| Cost metric | NCC | census | census | census |
| No. pyramid levels | 5 | 5 | 5 | 5 |
| Window size | 5–5–7–9–9 | $9 \times 7$ | $9 \times 7$ | $9 \times 7$ |
| No. path | n.a. | 8 | 8 | 8 |
| Penalty $P_1$ | n.a. | const. [1] | const. | const. |
| Penalty $P_2$ | n.a. | const. | $\max\left(\frac{P_2^0}{|I(p)-I(p-r)|}, P_1\right)$ | $\begin{cases} P_{21}, & C(p)=1 \\ P_{22}, & C(p)=0 \end{cases}$ |

[1] const. means constant value.

As shown in Table 2, we adopted a matching window size of $5 \times 5$ pixels for the top two epipolar pyramid levels, a $7 \times 7$ window size for the third level, and a $9 \times 9$ window size for the last two levels, as window size influences the performance of local matching methods. For SGM algorithms, a window size of $9 \times 7$ was adopted to calculate matching cost based on census transformation at every pyramid level. The number of cost aggregation paths in our SGM pipeline is eight. The only difference of the three SGM algorithms is the penalty function applied.

### 2.6. Stereo SAR Intersection

Once the disparities are exported after image matching, we can calculate the image coordinates of homologous points following the same epipolar rectification steps. The homologous points are then used for spatial intersection to obtain corresponding geographic coordinates. The range-Doppler (RD) model is used for the stereo intersection. The mathematical formulation of RD model is shown in Equation (10) where the first is Range Equation and the second is Doppler Equation.

$$
\begin{aligned}
|X_S - X_T| - c \cdot \frac{t_R}{2} &= 0 \\
\frac{V_S(X_T - X_S)}{|V_S||X_T - X_S|} &= \sin \alpha
\end{aligned}
\tag{10}
$$

$X_T$ is the geolocation of target at range time $t_R$, $X_S$ and $V_S$ are the geolocation and velocity vector of satellite sensor at azimuth time $t_A$, which is implicitly included in the second equation. The distance between sensor and target equals to the product of light velocity $c$ and range time $t_R$. In case of zero-Doppler, the squint angle $\alpha$ is zero.

Each point on the SAR image corresponds to a pair of range time $t_R$ and azimuth time $t_A$. For each pair of homologous points on the stereo images, we obtain two sets of RD model expressions, four equations in total. Therefore, the 3D geolocation of homologous points can be derived by solving the condition equations with least squares adjustment. After the stereo forward intersection, the dense geographic coordinates were resampled into a final gridded DSM. No post-processing or outlier detection is conducted during this process. The final DSMs are direct products produced from the disparity maps.

## 3. Results

In our experiments, we compared the Mount Song DSMs generated by SGM with three penalty functions and NCC method. All the radargrammetric DSMs were resampled

to a 10 m spatial resolution for height accuracy evaluation. The height accuracy was assessed in comparison with the reference airborne photogrammetric DSM. The height datum of radargrammetric DSMs and the reference airborne DSM is WGS84 ellipsoid, thus the SRTM DEM height was converted from EGM96 geoid to WGS84 datum for comparison. We processed epipolar images using stereo matching, spatial forward intersection, and triangulation interpolation to generate gridded DSMs. To genuinely reflect the matching performance of SGM, the disparities were directly put into the forward intersection step, and all the intersected spatial points were imported to the triangulation interpolation, no post-processing step or outlier detection was applied. The matching results and vertical accuracy assessment of the radargrammetric DSMs are presented in this section.

*3.1. Optimal Penalty Parameter Values*

Adopting the two-step "coarse-to-fine" parameter tuning strategy as presented in Section 2.4, we calculated the completeness of the disparity maps. The optimal parameter values for the three SGM penalty functions were determined using the disparity map with highest completeness. The optimal penalty parameter values for the Spotlight (SL) and Stripmap (SM) stereo pair are listed in Table 3.

**Table 3.** Optimal parameter values of SGM penalty functions.

| Matching Algorithm | $(P_1, P_2)$ SL Stereo Pair | $(P_1, P_2)$ SM Stereo Pair |
|---|---|---|
| SGM_const | (150, 200) | (150, 200) |
| SGM_gray | (150, 200) | (180, 250) |
| SGM_canny [1] | (150, 200) | (150, 200) |

[1] For canny penalty function, $P_{21}$ equals to $P_1$ and $P_2$ refers to $P_{22}$.

As illustrated by Table 3, the optimal parameter values depend on which penalty function was used. In terms of SGM_const and SGM_canny, we found the optimal parameter values of $P_1$ and $P_2$ are (150, 200) for both the SL and the SM stereo pair. With respect to SGM_gray, the optimal values for the SL data are the same as those for the other two functions; however, the parameter tuning for the SM data was more time-consuming and no regularity was found. These parameters provide an empirical reference to process other stereo SAR images with the SGM method. Meanwhile, the completeness of matching depends on which penalty function is applied.

*3.2. Matching Completeness*

Mismatched pixels on the epipolar image were marked as invalid disparity value on the disparity map. The mismatched pixels were detected by the disparity consistency checking in SGM pipeline and by NCC threshold judgement during NCC matching. Since the disparity values are directly used to generate the final DSMs without post-processing, the matching completeness was defined as the percentage of valid disparity values with respect to the total number of pixels of overlapping region in the epipolar image. The matching completeness of the two stereo pairs is listed in Table 4.

**Table 4.** Completeness of stereo matching methods.

| Matching Algorithm | SL Stereo Pair (SL_0925–SL_1001) | SM Stereo Pair (SM_0712–SM_0718) |
|---|---|---|
| SGM_const | 99.2% | 97.3% |
| SGM_gray | 98.6% | 96.1% |
| SGM_canny | 99.4% | 97.4% |
| NCC | 98.2% | 99.1% |

Table 4 shows that the completeness of all the SGM-derived disparity maps was higher than 96%. For NCC matching, we selected the NCC threshold with the highest DSM

precision after validation with the reference DSM and calculated the matching completeness. The disparity map completeness for each SGM algorithm was very close, but the height accuracy of DSM generated by each algorithm is different, as will be revealed in the next sub-section.

### 3.3. Quality Assessment of Radargrammetric DSMs

For each stereo matching algorithm, we obtained two gridded DSMs: one from the Spotlight pair and one from the Stripmap pair. The DSM reconstructed from the Stripmap pair using the SGM_canny algorithm is displayed with hillshading in Figure 4: the landform and terrain relief was clearly rebuilt by the SGM method. Visually, there is no distinct noise or outlier except a small area rendered as black spots in the northwest. This is consistent with the area obscured by rain clouds seen in image SM_0718. We selected three sub-regions covered by the airborne photogrammetric DSM, marked by blue, yellow, and orange rectangles in Figure 4, to quantitatively assess the quality of the DSM.

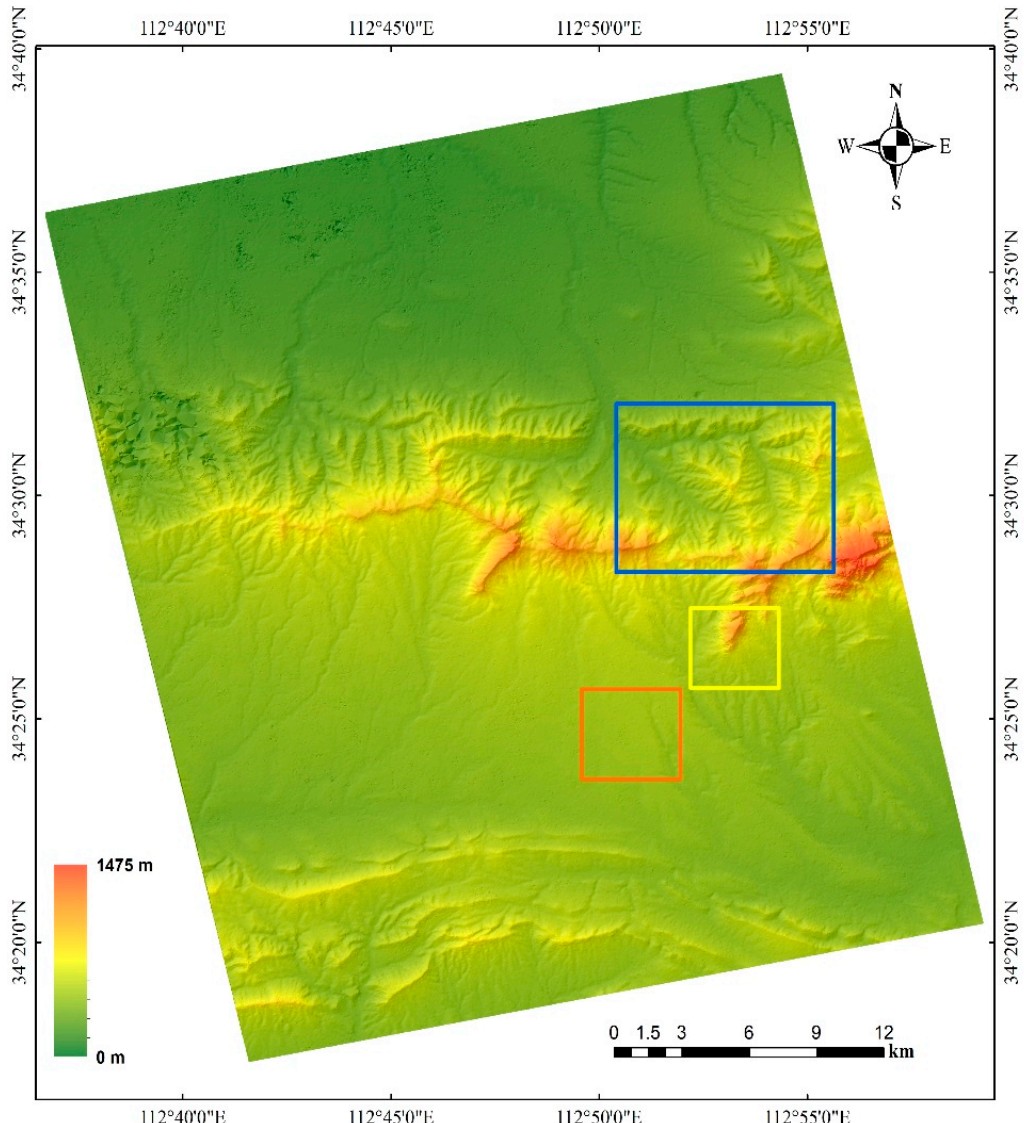

**Figure 4.** DSM with hillshading produced by SGM_canny algorithm from the Stripmap pair.

We subtracted the height of airborne reference DSM from the radargrammetric DSMs in the quantitative evaluation. Based on the derivation from reference height, several commonly used quality indicators were calculated to demonstrate and compare the accuracy of the radargrammetric DSMs. These metrics were mean error, mean absolute error (MAE),

the root mean square error (RMSE), and the linear error of 90% (LE90). LE90 is the absolute height error located where 90% of the sorted absolute height errors are smaller than or equal to it. Unlike the first three metrics, LE90 is a more reliable and robust indicator even when height errors do not obey normal distribution. In the following sub-sections, we sum up the results and assess the height accuracy for each test area.

### 3.3.1. Height Accuracy of Test Area 1

The mountainous test area 1 marked by the blue rectangle in Figure 4 is only within coverage of the SM data. The area has an East-West width of 8117 m and a North-South length of 7118 m. The height range is 260–1385 m. The DSM vertical accuracy indicators are summarized in Table 5.

**Table 5.** Height accuracy of the test area 1 in blue rectangle.

|                      | SRTM | NCC  | SGM_Const | SGM_Gray | SGM_Canny |
| -------------------- | ---- | ---- | --------- | -------- | --------- |
| Mean error (m)       | 3.0  | 1.8  | 2.1       | 2.1      | 1.7       |
| Mean Abs. error (m)  | 7.9  | 16.3 | 7.4       | 7.9      | 7.0       |
| RMSE (m)             | 10.9 | 22.5 | 9.9       | 10.5     | 9.5       |
| LE90 (m)             | 16.6 | 37.9 | 15.5      | 16.4     | 14.7      |

As shown, the indicator values from all the three SGM algorithms were smaller than those from the SRTM DEM. The LE90 from SGM_const and SGM_canny are more than 1 m smaller than that from SRTM DEM. The SGM_gray indicator values are slightly smaller than those from the SRTM DEM but the SGM_canny algorithm yielded the highest accuracy in terms of all accuracy indicators. The NCC method yielded the lowest accuracy. To demonstrate the results visually, all the DSM products for this test area are shown with hillshading in Figure 5.

In Figure 5, height is indicated by graduated colors. The figures coincide with the results presented in Table 5: Compared to the SRTM DEM, the SGM-derived DSMs reveal more details and more clarity. Despite the smoothness, topography of the SRTM DEM is more blurred than the SGM DSMs. The SGM DSMs show similar hillshading effects and can only be distinguished by the sporadic noise mainly situated on the top area of the figures. The massive amount of noise in the NCC DSM conceals the topographic details. To further reveal the differences of these height products, two sets of height profiles along the line segments marked as "m" ($lm$) and "n" ($ln$) in Figure 5c were extracted and plotted in Figures 6 and 7. We first interpret the profiles along $lm$ in Figure 6.

As seen in Figure 6a, the range of the height profile along $lm$ is 440–840 m. The height curves of SGM DSMs and SRTM DEM are close to each other. To examine the difference between the height curves more closely, we zoomed in on a portion marked by a dotted box in Figure 6b. The zoomed view shows the height profile from SGM_canny algorithm was closest to the reference DSM. The SRTM DEM profile generally deviates more from the reference DSM than the SGM profiles. NCC curve contains the biggest deviations in height. Profiles along $ln$ show a similar trend in Figure 7.

As shown in Figure 7a, the height range of the height profile along $ln$ is 350–1200 m. The height profiles of SGM DSMs and the SRTM DEM almost coincide. The zoomed view in Figure 7b shows the profile from SGM_canny DSM approaches the reference DSM most. The SRTM DEM profile contains larger height deviations than the SGM ones. The NCC curve deviates most from the reference DSM.

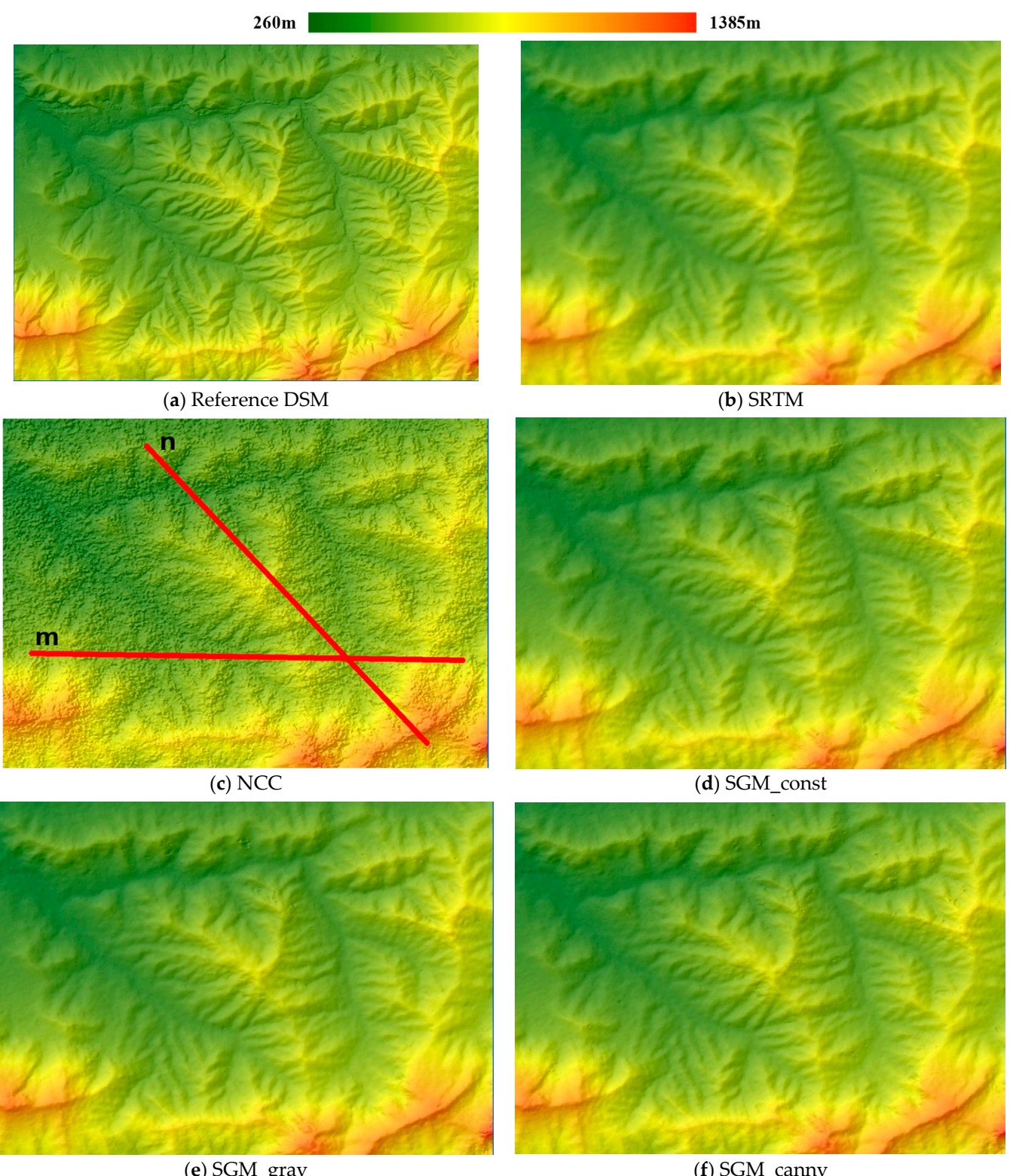

**Figure 5.** Hillshade of test area 1: (**a,b**) Hillshade of the reference photogrammetry DSM, and 30 m resolution SRTM DEM; (**c–f**) Hillshade of 10 m resolution radargrammetric DSMs produced from SM images by NCC, SGM_const, SGM_gray, and SGM_canny.

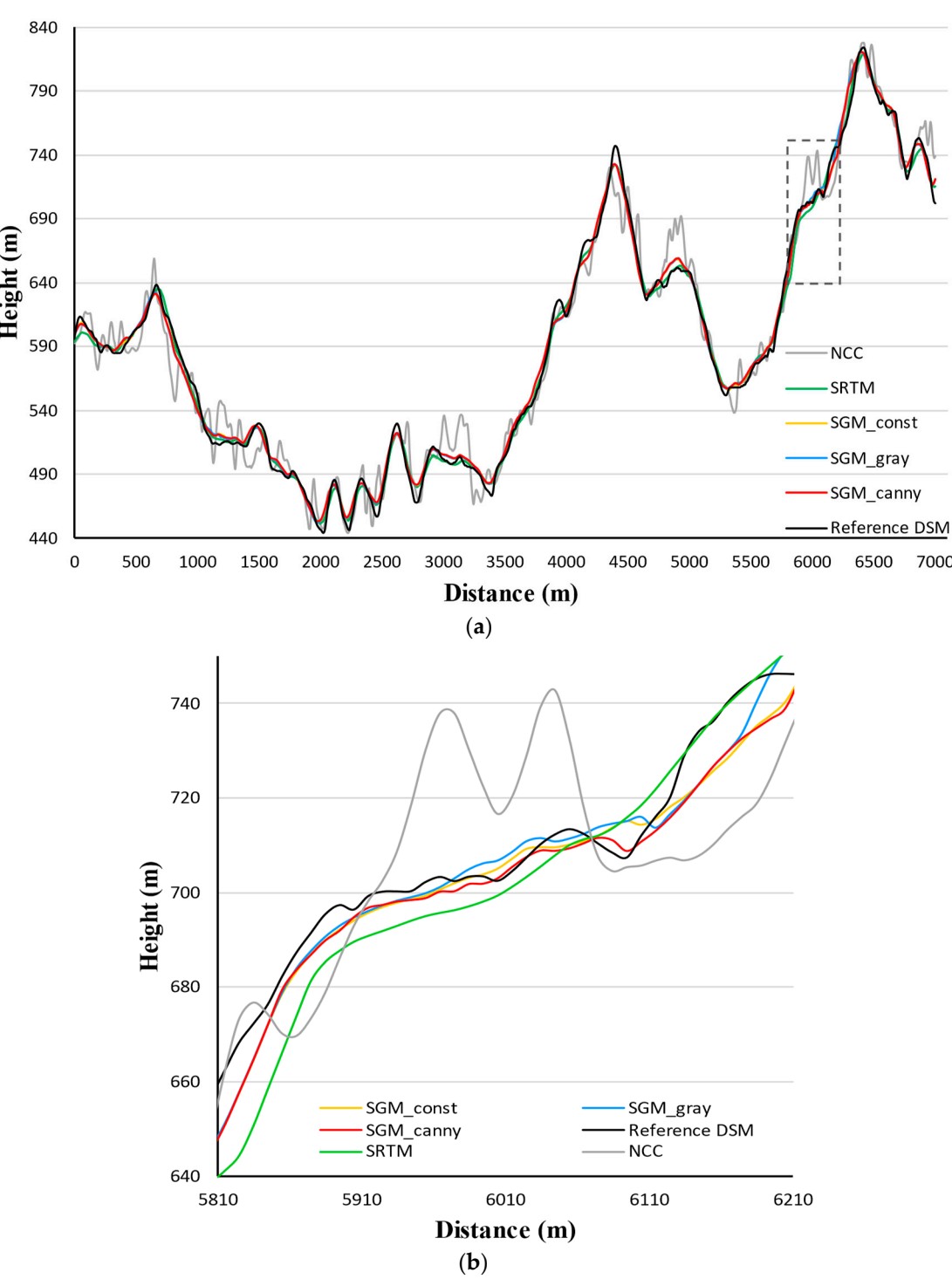

**Figure 6.** (**a**) DSM profiles along the line segment "m" in Figure 5c; (**b**) Zoomed view of the profiles corresponding to the dotted box in Figure 6a.

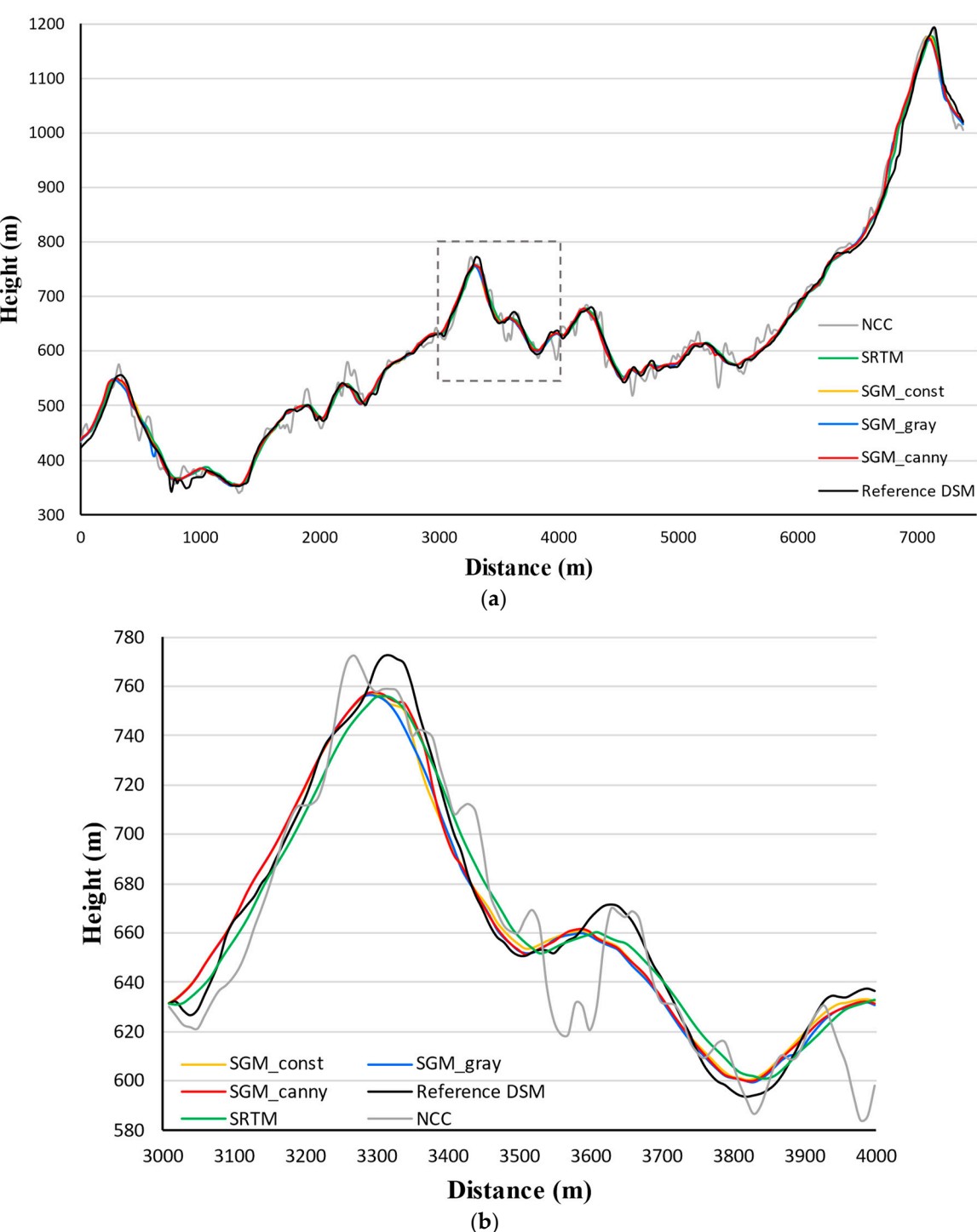

**Figure 7.** (**a**) DSM profiles along the line segment "n" in Figure 5c; (**b**) Zoomed view of profiles corresponding to the dotted box in Figure 7a.

To sum up, the two sets of height curves in Figures 6 and 7 follow similar trends. The length of both *lm* and *ln* is approximately 7000 m long. Along the entire distance, height profiles from SGM DSMs show no systematic offset compared to the reference DSM profile. The zoomed views reveal that profiles produced using the SGM_canny DSM most closely matches the reference height. The deviation distribution of each profile verifies the accuracy indicator values.

### 3.3.2. Height Accuracy of Test Area 2

The mountainous test area 2, marked with a yellow rectangle in Figure 4, is within the coverage of both the SM and SL data. It has an East-West width of 3313 m and a North-South length of 3368 m. The height range of this area is 440–1200 m. The accuracy indicators of DSM generated from Spotlight (SL) stereo pair and Stripmap pair are summarized in Tables 6 and 7.

**Table 6.** Height accuracy of DSM generated from Spotlight stereo pair in test area 2.

| SL Mode | SRTM | NCC | SGM_Const | SGM_Gray | SGM_Canny |
|---|---|---|---|---|---|
| Mean error (m) | −0.1 | 1.5 | 0.2 | 0.1 | 0.2 |
| Mean Abs. error (m) | 6.9 | 15.5 | 7.3 | 7.1 | 7.3 |
| RMSE (m) | 10.7 | 23.5 | 11.6 | 11.3 | 11.4 |
| LE90 (m) | 16.2 | 37.2 | 17.8 | 17.9 | 17.6 |

**Table 7.** Height accuracy of DSM generated from Stripmap stereo pair in test area 2.

| SM Mode | SRTM | NCC | SGM_Const | SGM_Gray | SGM_Canny |
|---|---|---|---|---|---|
| Mean error (m) | −0.1 | −0.9 | 0.1 | −0.5 | −0.2 |
| Mean Abs. error (m) | 6.9 | 17.7 | 6.1 | 6.7 | 5.6 |
| RMSE (m) | 10.7 | 24.4 | 9.6 | 10.5 | 8.9 |
| LE90 (m) | 16.2 | 40.2 | 13.8 | 15.4 | 12.5 |

Table 6 shows that for the SL data, the vertical accuracy of SGM DSMs is slightly lower than that of the SRTM DEM: the RMSE values range from 11.3–11.6 m and the LE90 values range from 17.6–17.9 m, 0.6–1.7 m larger than those from the SRTM DEM. For the SM data in Table 7, the vertical accuracy of SGM DSMs is the highest: the RMSE values range from 8.9 to 10.5 m, and the LE90 values range from 12.5 to 15.4 m which are generally 2 m smaller than those from the SRTM DEM. The NCC method provides indicator values about two times larger than SRTM. All in all, the SGM_canny algorithm provides the highest accuracy with the SM mode stereo pair. The DSM products for test area 2 are shown in Figure 8 with hillshading.

In Figure 8, SGM DSMs reveal more terrain details and more clarity than the SRTM DEM. Again, the NCC DSMs are full of noise. Comparing the Spotlight DSMs in the second row and Stripmap DSMs in the third row, we found the results show similar smoothness. However, notice the region marked with the red ellipse in Figure 8b, the hillshading appears as triangulation facets. The triangulation facets also appear in all the SL mode SGM DSMs in Figure 8d–f. Only the SGM_canny result in Figure 8f shows less facets and more complete mountain terrains than the other Spotlight results. These facets indicate that stereo matching in this area found very low density of correspondences. Thus, the hillshading appeared as triangulation facets after the triangulation interpolation. To investigate the causes, the Spotlight stereo amplitude images, the epipolar images and the disparity map in this region are reviewed and shown in Figure 9.

Figure 9a shows the subsets of ascending SL mode amplitude images corresponding to the facets, and a distinct layover can be seen on the slope facing towards the radar sensor. The geometric distortion is considerably corrected after epipolar rectification as shown in Figure 9b. However, there is no texture information in the severe layover region which appears as white color on the rectified epipolar images. In Figure 9c, the subset of the Spotlight mode disparity map generated by the SGM_gray algorithm indicates invalid disparity values by white-colored regions appearing in the layover area. Homologous points were determined by the disparities and intersected to derive 3D geographic points. We plotted the geographic points with black color overlaying on the reference DSM hillshade. The reference DSM is almost completely covered by the dense geographic point layer but no points exist where layover effects appear. Therefore, facets were produced after the triangulation interpolation and remained in the final gridded DSM. In turn, the layover impact also explains why the

height accuracy of Spotlight results in Table 6 is lower than that of Stripmap results in Table 7. For the same area in the descending SM data, there are no layover distortions, so that the height was built successfully without any triangulation facets appearing. As we generated epipolar images and final DSMs for both SM and SL data at the same 10 m resolution, the layover impact is the main reason for the difference in accuracy.

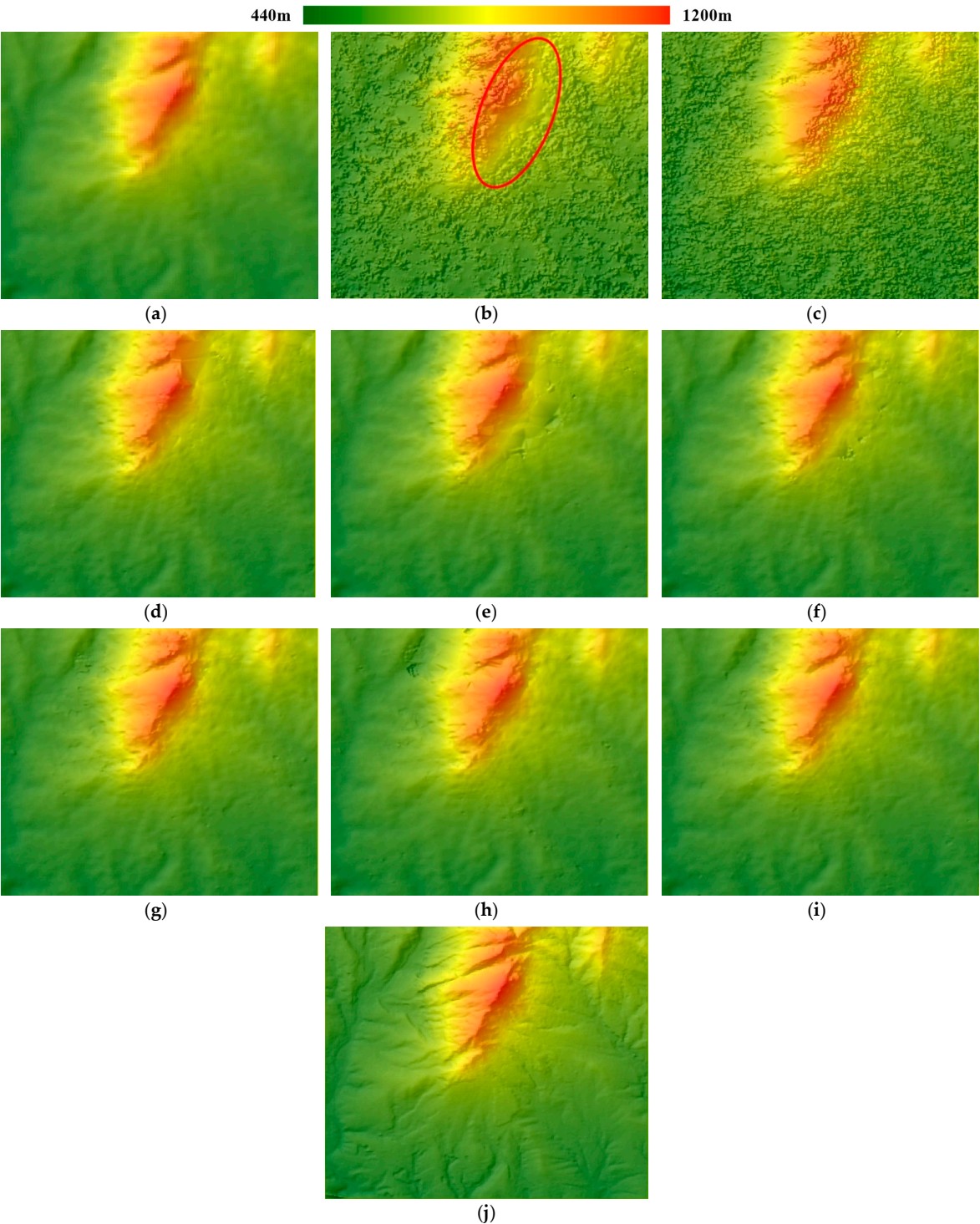

**Figure 8.** Hillshade of test area 2: (**a**–**c**) Hillshade of SRTM DEM, SL mode DSM by NCC, and SM mode DSM by NCC; (**d**–**f**) Hillshade of SL mode DSMs by SGM_const, SGM_gray, and SGM_canny; (**g**–**i**) Hillshade of SM mode DSMs by SGM_const, SGM_gray, and SGM_canny, and (**j**) Hillshade of reference DSM.

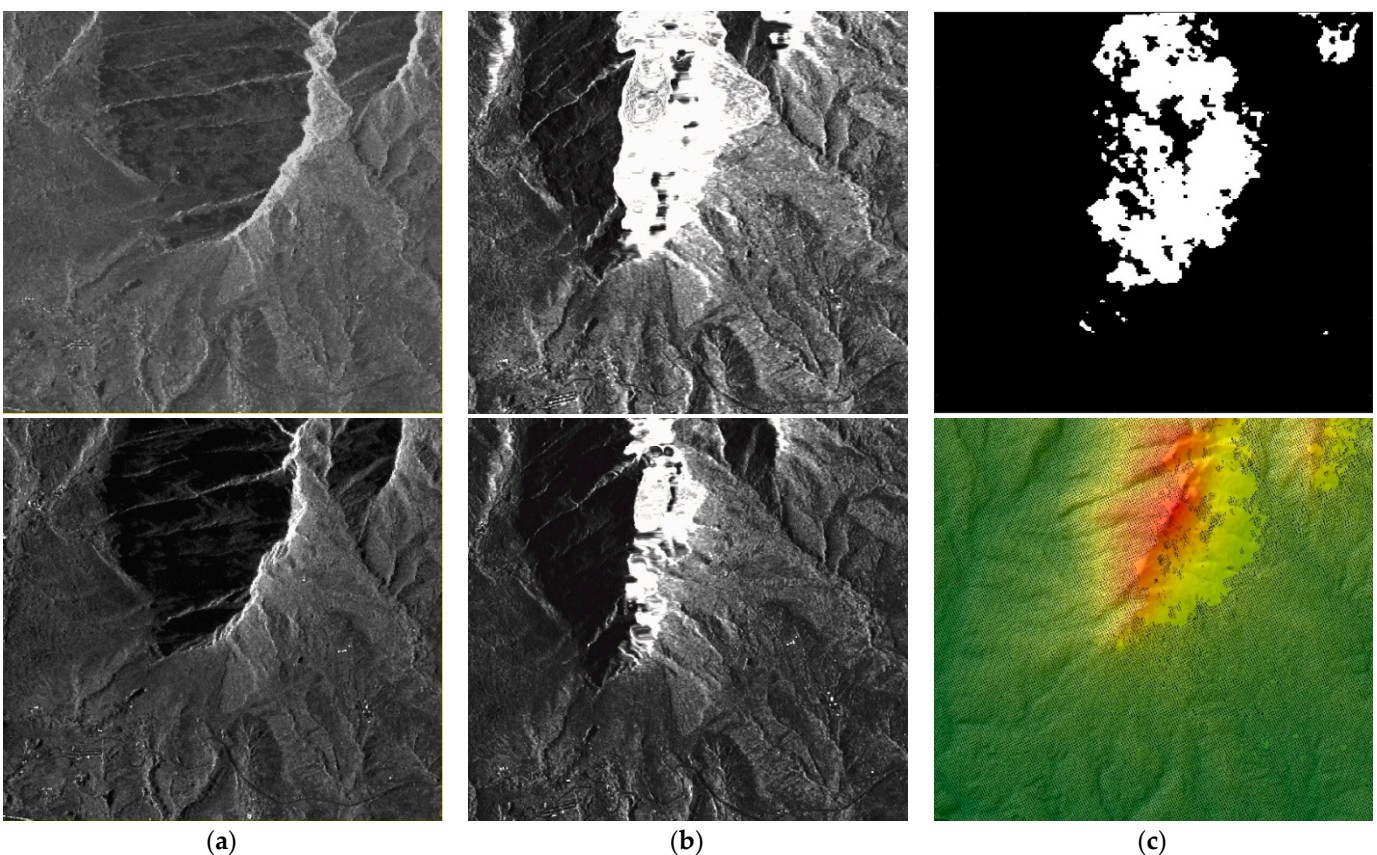

**Figure 9.** (**a**) Amplitude image subset of SL_0925 and SL_1001; (**b**) Epipolar image subset of SL_0925 and SL_1001; (**c**) subset of the Spotlight mode disparity map derived by SGM_gray (Black colors represent valid disparity values; White color represents invalid disparity values); Geographic points generated from the disparity map overlaying on the reference DSM.

### 3.3.3. Height Accuracy of Test Area 3

Test area 3, a flat area marked with an orange rectangle in Figure 4, is within the coverage of both the SM and SL images. It has an East-West width of 3705 m and a North-South length of 3797 m. The height range is 410–525 m. The accuracy indicator values of DSM generated from Spotlight (SL) stereo pair and from Stripmap (SM) stereo pair are summarized in Tables 8 and 9.

**Table 8.** Height accuracy of DSM generated from Spotlight stereo pair in test area 3.

| SL Mode | SRTM | NCC | SGM_Const | SGM_Gray | SGM_Canny |
|---|---|---|---|---|---|
| Mean error (m) | 0.7 | 1.4 | 1.2 | 1.4 | 1.5 |
| Mean Abs. error (m) | 1.5 | 15.2 | 1.9 | 2.0 | 2.1 |
| RMSE (m) | 1.9 | 21.0 | 2.1 | 2.1 | 2.2 |
| LE90 (m) | 3.3 | 38.6 | 3.9 | 3.9 | 4.1 |

**Table 9.** Height accuracy of DSM generated from Stripmap stereo pair in test area 3.

| SM Mode | SRTM | NCC | SGM_Const | SGM_Gray | SGM_Canny |
|---|---|---|---|---|---|
| Mean error (m) | 0.7 | 0.7 | 1.1 | 0.7 | 0.9 |
| Mean Abs. error (m) | 1.5 | 15.9 | 1.8 | 1.8 | 1.7 |
| RMSE (m) | 1.9 | 20.8 | 2.1 | 2.3 | 2.0 |
| LE90 (m) | 3.3 | 36.6 | 3.9 | 3.8 | 3.6 |

Tables 8 and 9 reveal that for both the SL data and SM data, SGM method yields accuracy values that are quite close to the SRTM DEM. The RMSE values and LE90 values from SGM results are only several decimeters larger than those from the SRTM DEM. In addition, the accuracy indicators of SGM_const, SGM_gray, and SGM_canny results are very close, almost identical. Still, NCC delivers the lowest accuracy. In Figure 10, we plotted the DSM products for this test area with hillshading.

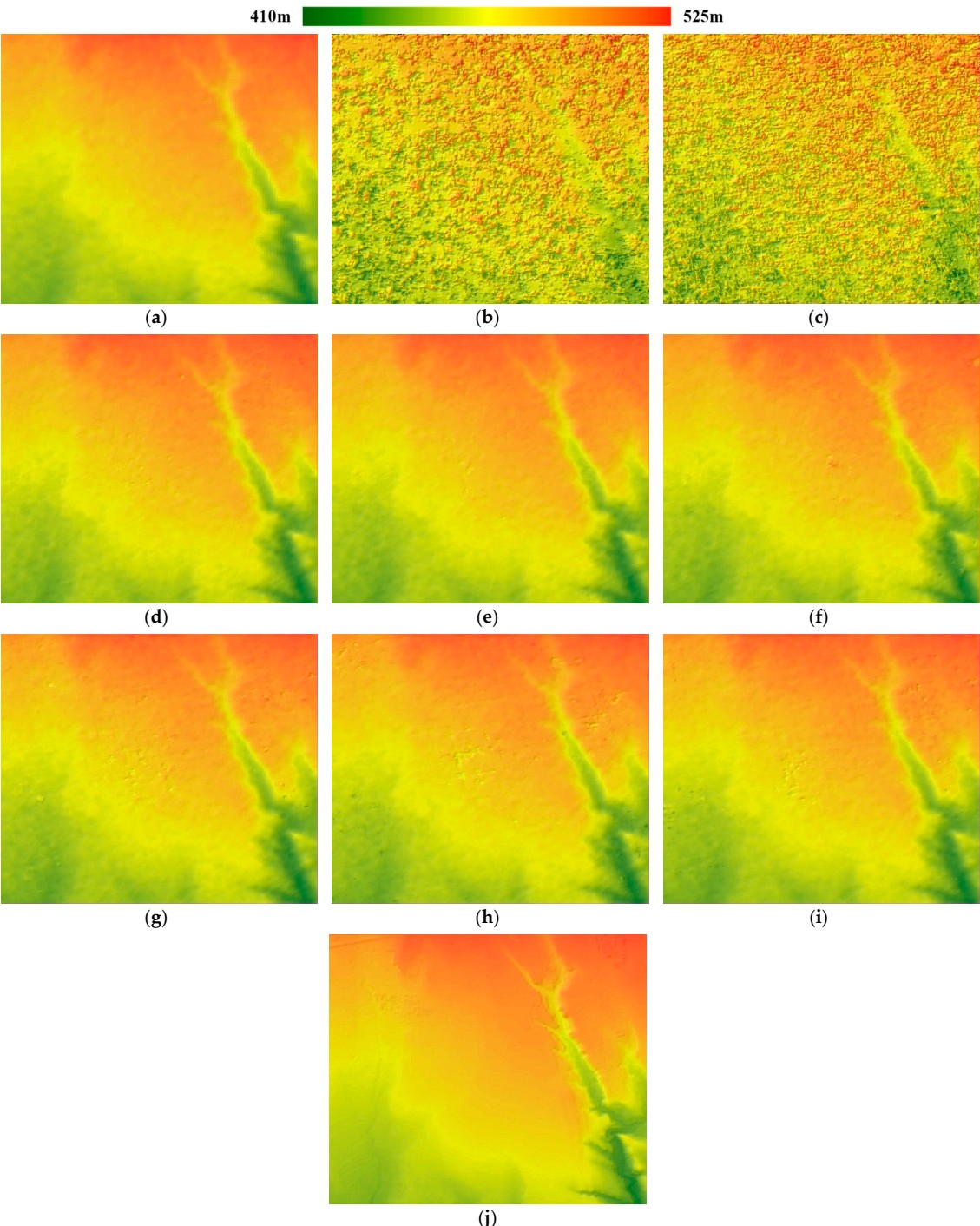

**Figure 10.** Hillshade of test area 3: (**a–c**) Hillshade of SRTM DEM, SL mode DSM by NCC, and SM mode DSM by NCC; (**d–f**) Hillshade of SL mode DSMs by SGM_const, SGM_gray, and SGM_canny; (**g–i**) Hillshade of SM mode DSMs by SGM_const, SGM_gray, and SGM_canny, and (**j**) Hillshade of reference DSM.

Figure 10 shows that NCC DSMs are too noisy to reflect small topographic changes, while the SGM DSMs reconstruct detailed topography in this area. With the same 10 m resolution, the SL mode SGM DSMs are less noisy than the SM mode DSMs, though the corresponding quality metric values visible in Table 8 are very close to those in Table 9. The SGM algorithms produced quite similar results for the Spotlight data, while for the Stripmap data, the SGM_canny algorithm generated the smoothest result with the least noise in Figure 10i.

### 3.4. Matching Processing Time

We recorded the processing time of each radargrammetric matching algorithm to evaluate the efficiency. All radargrammetric algorithms were implemented with C++ language and ran on a notebook computer with an Intel Core i5-7300HQ CPU (4 cores, 2.5 GHz). The time consumption is summarized in Table 10.

**Table 10.** Matching processing time of all stereo matching algorithms used in this study.

| Time (s) | SL Epipolar Images (1650 × 1689 Pixels) | SM Epipolar Images (5902 × 5261 Pixels) |
|---|---|---|
| SGM_const | 5.5 | 32.2 |
| SGM_gray | 5.7 | 52.2 |
| SGM_canny | 5.8 | 48.5 |
| NCC | 27.6 | 820.6 |

The matching for SL stereo pair took a shorter time than the SM stereo pair, since the size of SL mode epipolar image is one third of that of the SM mode epipolar image. For SL data, it cost less than six seconds for all SGM algorithms and 27.6 s for the NCC method. For SM data, all SGM algorithms spent less than 1 min, while NCC spent more than 13 min. Notice that SGM_const is the fastest for using constant penalty values, but SGM_gray and SGM_canny cost only a few more seconds.

## 4. Discussion

### 4.1. Evaluation of Stereo Matching Methods

Our results show that the DSMs generated from NCC method are of much lower quality than SGM method and the SRTM DEM, while the height accuracy of SGM DSM results without layover is higher than the SRTM DEM in mountainous areas and comparable in flat areas. Specifically, the hierarchical NCC method yielded RMSE values of 22.5–24.4 m and LE90 values of 37.2–0.2 m in mountainous areas, with RMSE values of 20.8–21 m and LE90 values of 36.6–38.6 m in flat areas. The hierarchical SGM method provides RMSE values of 8.9–11.6 m and LE90 values of 12.5–17.9 m in mountainous areas, with RMSE values of 2.0–2.3 m and LE90 values of 3.6–4.1 m in flat areas.

Previous studies have used the same test data as this study to generate stereo DSMs [19,21]. We selected almost identical test areas as the two previous studies to make the height accuracy comparable. A direct radargrammetric method adopting NCC to search for the optimal height in object space was proposed in [19]. An adaptive-window least squares matching (LSM) method was proposed in [21] and compared with the direct method. The direct method in [19] yielded a vertical accuracy at the same level as the hierarchical NCC method in this study. The least squares matching delivered RMSE values of 9.4–13.8 m and LE90 values of 16.0–23.5 m in mountainous areas, with RMSE values of 2.0–5.2 m and LE90 values of 3.3–8.8 m in the flat area. According to these indicator values, our SGM pipeline produced DSM with the highest vertical accuracy, the SRTM DEM was suboptimal, the LSM-derived DSMs were less accurate than the SRTM DEM, and NCC methods yielded the lowest vertical accuracy.

NCC is a typical local matching method which implicitly assumes disparities of all pixels within the matching window are the same. In addition, the obtained disparity for each pixel is just locally optimized. Thus, the matching accuracy is limited and post-

processing is required to filter outliers [36]. LSM is also a local matching method but applies an affine transformation between correspondences and compensates for radiometric distortion to improve matching accuracy. However, LSM requires initial values usually provided by NCC matching and performs an iteration procedure on every pixel. LSM is too sensitive to converge to a robust solution as well as too time-consuming. SGM, however, takes the disparity difference of adjacent pixels into consideration and integrates regularization constraints in the algorithm, robustly preserving the true disparities. Our results demonstrated that even in a layover area lacking texture information, the vertical accuracy of SGM DSMs is close to the SRTM DEM, and the SGM pipeline ran much faster than the NCC matching algorithm, with the SGM method not only ensuring matching accuracy but also increasing efficiency.

### 4.2. DSM Accuracy Difference between Mountainous Areas and Flat Areas

Our results show the mean Abs. error, RMSE, and the LE90 values in mountainous areas are always larger than those in flat areas for both SL mode and SM mode images. In the mountainous test areas, the LE90 values of SGM DSMs are all less than 18 m. In the flat topography, the LE90 values of SGM DSMs were less than 4 m. SRTM DEM and NCC DSM also have larger height errors in mountainous test sites. The fact that height residuals are larger in mountainous areas than in flat areas was also observed in many previous studies including radargrammetric approaches [19–21,25] and InSAR DEMs [42–44]. This is because SAR images often contain geometric distortions including layover and shadow in high-relief terrains [3], and the geometric distortions result in data gaps affecting both radargrammetric terrain mapping and InSAR mapping. Therefore, the height reconstruction in mountains is more difficult and less accurate than in plains.

Our results also show how layover in mountainous areas influenced the height reconstruction. Epipolar images in [25] demonstrate the same impacts of layover in the mountainous region and study [45] shows similar triangulation facets resulting from layover in InSAR DEMs. Missing topographic information caused by radar layovers leads to matching failure in radargrammetry and incorrect phase unwrapping in InSAR processing. Nevertheless, the height accuracy statistics of our SL mode SGM DSMs is only slightly lower than those from the SRTM DEM. The triangulation facets could be improved by fusing DSMs generated from ascending and descending stereo image pairs.

### 4.3. On the Influence of Different Penalty Functions for $P_2$

In terms of the three penalty functions for the calculation of $P_2$, the height accuracy differences from three penalty functions are very small in flat test areas; while in the mountainous areas, for the Stripmap stereo pair, SGM_canny presents the smallest accuracy indicator values. The height profiles and hillshades verify the accuracy values: the profiles of SGM_canny DSM have the minimum residuals to the reference height profile and the SGM_canny algorithm yields the smoothest hillshades with the least noise. For the Spotlight stereo pair, the accuracy indicator values from three penalty functions are almost identical, but the hillshades in the layover area demonstrate that SGM_canny yields more complete mountain terrains and less facets than the other two algorithms.

The gray gradient penalty function is usually used in SGM algorithms for optical images [12,46]. Previous study [47] compared different penalty functions of SGM on optical image matching, and inferred that the gray gradient function outperforms the constant penalty function. Our results show, for the Stripmap data, the SGM_const algorithm using the constant penalty function delivers higher accuracy than the SGM_gray algorithm in mountainous test areas; for the Spotlight data, the SGM_gray algorithm yields more triangulation facets in the layover area than SGM_const algorithm. This may be attributed to the speckles and noise inherent in SAR images. The gray gradient calculation is sensitive to speckles and noise that introduce gross error and affect the matching accuracy. According to the results of our test data, we recommend adopting canny penalty function in SGM for stereo SAR image matching.

## 5. Conclusions

In this paper, we implemented a hierarchical SGM dense matching pipeline for stereo SAR image matching, which can produce high quality radargrammetric DSMs in forested mountain areas. The pipeline is designed in a user-friendly manner with only two user-set penalty parameter values. The output disparity maps can be directly used to generate high-quality DSMs without post-processing. The SGM-derived DSMs, NCC-derived DSMs, and the 30 m resolution SRTM DEM were analyzed in terms of accuracy and efficiency. The results show that in mountainous regions, the height accuracy of the SGM-derived DSM is higher than that of the SRTM DEM; in flat regions, the height accuracy of the SGM-derived DSM is at the same level as the SRTM DEM. The NCC matching method yielded height accuracy twice that of the SRTM DEM. Our SGM dense matching pipeline provides the highest vertical accuracy and processing efficiency. For severe layover areas, the number of successfully matched homologous points is inadequate as the texture information is insufficient, but the height accuracy of the SGM-derived DSMs is still close to the SRTM DEM owing to the regularization constraints. Furthermore, three penalty functions were evaluated in our SGM matching pipeline. The penalty function of the canny edge detector yields the highest vertical accuracy. In the future, efforts will be devoted to the development of a new penalty function more suitable for SAR image matching. Additionally, different matching costs such as census, NCC, and mutual information will be evaluated. Stereo DSMs generated from different orbits will be fused to improve the height accuracy in layover areas.

**Author Contributions:** Conceptualization, J.W. and T.B.; methodology, J.W.; software, J.W.; validation, J.W. and K.G.; formal analysis, J.W.; investigation, J.W.; resources, M.L. and T.B.; data curation, L.Z. and M.L.; writing—original draft preparation, J.W.; writing—review and editing, T.B., N.H., and U.S.; visualization, K.G. and J.W.; supervision, U.S. and N.H. All authors have read and agreed to the published version of the manuscript.

**Funding:** This research was supported by the National Natural Science Foundation of China (NSFC) under grant numbers 41271457 and 41774006.

**Acknowledgments:** We would like to thank the German Aerospace Center (DLR) for providing the test datasets via the DLR AO LAN0793, LAN0634, and LAN2245. We appreciate Miaozhong Xu from LIESMARS for providing the airborne photogrammetric DSM in the Mount Song area.

**Conflicts of Interest:** The authors declare no conflict of interest.

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
