# Peer review of "Radargrammetric DSM Generation by Semi-Global Matching and Evaluation of Penalty Functions"

_remotesensing, doi:10.3390/rs14081778_

Round 1

Reviewer 1 Report

The paper investigates and demonstrate in a rather clear way the perfromance of Semi Global Matching based DEM reconstruction methods for applied to stereo SAR data.

I am not a full expert on this and not fully aware on all recent developments in this field, but a quick rechereche quickly discovered recent excellent papers on this topic that should be referenced, e.g. Bagheri et al., A framework for SAR-optical stereogrammetry over urban areas doi: 10.1016/j.isprsjprs.2018.10.003.

The paper is well written, the research design clear and thre results are predominantly convincing.

I have a few comments for a revision of this paper:

p 5, l 186: "radargrammetic DSMs" is misleading assuming you compare your DSM with something different. Please clarifiy that SRTM is interferometric and your DEM is radargrammetric.

p 5, l 189: " ... that also applies the case.". Please check syntax and semantics. I don't understand what you want so say here.

p 5, l 197: You write that the SRTM DEM was used to precondition the data to a large extent, i.e. the required disparity search range was very small. At the end you even compare your results with SRTM-data! You should describe what would happen if such data were not available.

P l 204: "colors of blue and red ...." is misleading. Those colors (red and blue/green?) represent the individual images and any changes caused by disparity or temporal changes become visible. There is no synthetic pseudo color involved. You might explain how white color  can occur if only read and green are mixed. Or did you mix one image to red and green channels? Please clarify preciesely.

p 8, l 254: I don't understand why the search range is adaptive because in l. 259 you write that the search range is +-4. Adaptive would mean you adjust the search range, e.g. on the quality of the result.

p 9: I know the normalized cross correlation very well, but I don't understand equation 9. Usually the shift parameters p,q appear as shifts in one of the windows. It is not clear what you mean with center pixel (p,q) on both of the matching windows. The shift must be mentioned somewhere.

p 10, eq 10: is wrong. Radar range measurement is (c * t sub R) / 2

p 12, chapter 3.3: Any DEM reconstruction method must consider the achievable resolution because it is very easy to reduce the vertical error at the cost of horizontal resultion. The manuscript is not addressing this fact quantiatively.

p 14, l 446: check typesetting of "lm exponent lm"

p 21, l 538: In several of your tables it is visible that spotlight data does not produce better results than stripmap data as would be expected. This fact must be discussed and also in combination with the achievable resolution. E.g. I would expect that at full resolution the SL data gain in vertical accuray is approximately proportional to the increased resolution beacuse of finer disparity measurements. If targeting at the same resolution of the output DEM, SL data should gain additionally from the increased window size in pixels. Etc etc.

Author Response

Thank you very much for your careful reading and suggestive comments. The questions and suggestions has been listed with my answers in the following:

  • I am not a full expert on this and not fully aware on all recent developments in this field, but a quick rechereche quickly discovered recent excellent papers on this topic that should be referenced, e.g. Bagheri et al., A framework for SAR-optical stereogrammetry over urban areas DOI: 10.1016/j.isprsjprs.2018.10.003.

As suggested, the new reference is added as you can see in line 800-801 of the updated version in the manuscript, and a sentence is added as you can see in line 131-135 of the updated version in the manuscript: “In 2018, Bagheri et. al. proposed a semi-global matching framework to investigate the potential of 3D reconstruction from SAR-optical image pairs and feasibility of generating point clouds with a median accuracy [32], but the framework is designed for extracting sparse homologous points from SAR-optical stereo pair and not for DSM reconstruction.”

  • p 5, l 186: "radargrammetric DSMs" is misleading assuming you compare your DSM with something different. Please clarify that SRTM is interferometric and your DEM is radargrammetric.

This sentence is revised to “NASA’s 30 m spatial resolution SRTM DEM [2] acquired with InSAR approach was also used for the height accuracy comparisons with the DSMs generated with our radargrammetric methods.” as you can see in line 205-207 of the updated version in the manuscript.

  • p 5, l 189: " ... that also applies the case.". Please check syntax and semantics. I don't understand what you want so say here.

This sentence is revised to “The SRTM DEM and the radargrammetric DSMs are comparable because they were generated from short wavelength SAR images, thus the extracted heights all refer to the upper surface of objects on earth.” as you can see in line 207-209 of the updated version in the manuscript.

  • p 5, l 197: You write that the SRTM DEM was used to precondition the data to a large extent, i.e. the required disparity search range was very small. At the end you even compare your results with SRTM-data! You should describe what would happen if such data were not available.

As suggested, a description is added as you can see in line 231-234 of the updated version in the manuscript:” When such a-priori DEMs are not available, we can generate them with commonly-used radargrammetric methods such as NCC matching, or simply by applying an identical mean height value as proposed by [23].”

  • P l 204: "colors of blue and red ...." is misleading. Those colors (red and blue/green?) represent the individual images and any changes caused by disparity or temporal changes become visible. There is no synthetic pseudo color involved. You might explain how white color can occur if only read and green are mixed. Or did you mix one image to red and green channels? Please clarify precisely.

Thank you for pointing out and correcting this problem. Yes, there is no synthetic pseudo color involved. And the white regions refer to regions that lack matching texture. The sentence is revised to “The blue and red colors represent the disparity between the left and right epipolar images. Only one color showing in the overlapping areas, indicates that there is no disparity between the homologous points, and most of the disparities were eliminated in each epipolar pair. The white regions refer to areas that lack matching texture. This will be shown in section 3.3.2.” as you can see in line 240-245 of the updated version in the manuscript.

  • p 8, l 254: I don't understand why the search range is adaptive because in l. 259 you write that the search range is +-4. Adaptive would mean you adjust the search range, e.g. on the quality of the result.

Thank you for this good question. Unlike the SGBM method in OpenCV, which assigns a fixed disparity search range such as [-32,32] or [-16,16] for all pixels, the search range in our pipeline is pixel-wise, which is determined by an initial disparity value  transmitted from each pixel  on a higher pyramid level. Therefore, each pixel has its own search range as [ ], rather than [-4, 4].

To make this part more readable and clearer, the sentence is revised to “Specifically, the disparity map of a higher level is upscaled by two to match the size of next level, so the disparity value of pixel  is multiplied by two to serve as the initial disparity value  for the same pixel on a lower level. The complete disparity search range for pixel  on the lower level is defined as [ ]. Thus, a pixel-wise disparity range is determined for each pixel on the lower level, and the radius of four guarantees that the range is not too large to increase ambiguities or too narrow to miss the correct disparity value.” as you can see in line 294-301 of the updated version in the manuscript.

  • p 9: I know the normalized cross correlation very well, but I don't understand equation 9. Usually the shift parameters p,q appear as shifts in one of the windows. It is not clear what you mean with center pixel (p,q) on both of the matching windows. The shift must be mentioned somewhere.

Thank you for this comment. Consistent with Equation 1-5,  and  represent the homologous pixel pair on base and matching image, instead of shift values. Since we are matching epipolar images, the shifts of homologous points are only along the epipolar direction, meaning that searching for homologous points is along the epipolar direction, too. Thus, for pixel  on the base image, its possible homologous point  on the matching image is , where  is the disparity (shift) along epipolar direction. There is no shift in the direction perpendicular to the epipolar lines.

As suggested, the sentence is revised to be “where  is the pixel on the base image for which the NCC coefficient is calculated,  is the pixel with disparity shift  along epipolar direction on the matching image: ;  represents the matching window on the base image and the matching image;” as you can see in line 361-364 of the updated version in the manuscript.

  • p 10, eq 10: is wrong. Radar range measurement is (c * t sub R) / 2

As suggested, the Equation 10 has been corrected as you can see in line 389 of the updated version in the manuscript.

  • p 14, l 446: check typesetting of "lm exponent lm"

As suggested, the sentence is revised to be “We first interpret the profiles along  in Figure 6.” as you can see in line 492 of the updated version in the manuscript.

  • p 12, chapter 3.3: Any DEM reconstruction method must consider the achievable resolution because it is very easy to reduce the vertical error at the cost of horizontal resultion. The manuscript is not addressing this fact quantitatively.
  • p 21, l 538: In several of your tables it is visible that spotlight data does not produce better results than stripmap data as would be expected. This fact must be discussed and also in combination with the achievable resolution. E.g. I would expect that at full resolution the SL data gain in vertical accuracy is approximately proportional to the increased resolution because of finer disparity measurements. If targeting at the same resolution of the output DEM, SL data should gain additionally from the increased window size in pixels. Etc etc.

I will answer these two questions together as they’re both concerned with the achievable resolution and related.

For stereo SAR image matching, the achievable resolution of final radargrammetric DSM is dependent on the number of multi-looking. Multi-looking and filtering should be applied to the SAR amplitude images, otherwise the speckle and noise in the SAR images will strongly affect the matching performance and introduce many outliers in final DSM.  For our test data, we aim at generating DSMs with the same resolution. Therefore, we used  multilooking for the spotlight images and 3 3 multi-looking for the stripmap images. Based on the multi-looking, the achievable resolution is of the final DSM is 10 m.

As suggested, sentences are added in line 221-229 of the updated version in the manuscript: “For the Stripmap mode images with the range resolution of 1.8m and the azimuth resolution of 3.3 m, we performed 3 3 multi-looking on the amplitude images. For the Spotlight mode images with the range resolution of 1.2m and the azimuth resolution of 1.6 m, we performed 5 5 multi-looking on the amplitude images. Therefore, the epipolar images of both stereo pairs were generated with 10 m spatial resolution, and DSMs were de-rived from both stereo pairs with the same resolution of 10 m. If amplitude images at full resolution are used to generate epipolar images, the matching process will introduce many mismatched disparities or voids due to the speckles and noise maintained in SAR images.”

As to the reason why spotlight data does not produce better results than stripmap data, I will discuss the question in mountainous areas and flat areas separately. For mountainous areas, i.e., test area 2 in section 3.3.2, Table 6(a) shows the height accuracy of Spotlight data is not better than that of Stripmap data in Table 6(b). The reason is the layover impact on the Spotlight image decreases the height accuracy. Inadequate homologous points were found in the layover regions and the final DSM hillshade appears as facets. This phenomenon is seen in Figure 8 and Figure 9, and discussed in the paragraphs below the figures.

As suggested, sentence is added as you can see in line 565-571 of the updated version in the manuscript: “In turn, the layover impact also explains why the height accuracy of Spotlight results in Table 6(a) is lower than that of Stripmap results in Table 6(b). For the same area in the descending SM data, there are no layover distortions, so that the height was built successfully without any triangulation facets appearing. As we generated epipolar images and final DSMs for both SM and SL data at the same 10 m resolution, the layover impact is the main reason for the difference in accuracy.”

For the flat areas, i.e., test area 3 in section 3.3.3, Table 7(a) shows the height accuracy of Spotlight data is quite close to that of Stripmap data in Table 7(b). As we generated epipolar images and final DSMs for both SM and SL data at the same resolution of 10 m, these accuracy metric values are reasonable and within expectation.

As suggested, the sentence is revised to be “With the same 10 m resolution, the SL mode SGM DSMs are less noisy than the SM mode DSMs, though the corresponding quality metric values visible in Table 7(a) are very close to those in Table 7(b).” as you can see in line 590-592 of the updated version in the manuscript

Reviewer 2 Report

From the methodological point of view the paper is presented in a very attractive manner. My general comment is that the authors should consider improving the language and phrasing.

Comments to authors:
From the methodological point of view the paper is presented in a very attractive manner. My
general comment is that the authors should consider improving the language and phrasing.
Lines 13 – 15: “Stereo image matching in radargrammetry is the process of deter-13 mining
correspondences and the performance of image matching influences the final 14 quality of DSM used
for spatial-temporal analysis of landscapes and terrain”.
Please reformulate the phrase.
Line 16: “methods are most commonly-used but and usually produce”
Line 19: “To fill this gap, we apply applied a hierarchical”
Line 35: “Since the early 2000s, global SAR-based terrain models have been acquired from SAR
images imagery.”
Line 36: “For instance, SRTM and TanDEM-X DEM are the most representative global DEMs.”
Line 46: “given pair of images usually either InSAR or radargrammetry technique can…”
Line 54: “local matching algorithms and global matching algorithms.”
Line 69: “The dependence of windows implicitly assumes that disparities of all”
Line 113: “DATE workflow [26-28], which exploits the Semi-Global Block Matching (SGBM) algorithm”
Lines 116 – 117: “Whereas the workflow does not transfer disparity from the higher level to a lower
116 level to reduce the disparity search range for a lower-level matching, instead it converts 117 the
disparities to height correction values at every level”
Please reformulate the phrase.
Line 127: “where only two penalty parameter values need are needed/are necessary to be set.”
Please reformulate the phrase.
Line 133: “approach were also included to compare in the vertical precision comparation”
Line 134: “not only produces produce DSMs with higher”
Lines 137 – 138: “edge detector is more suitable than the constant penalty and the gray gradient
penalty 137 functions used in stereo SAR image matching.”
Please reformulate the phrase.
Lines 148 – 149: “based on the Range-Doppler model to derive the 3D geographic coordinates, and a
triangulation interpolation yields a final gridded DSM.” Please reformulate the phrase.
Line 150: “function of SGM as will be shown in Equation (1),”
Line 160: “from 150 m to 1512 m above sea level, Mount Song possesses is characterized by a
complex”
Line 161: “geological history of over 3.6 billion years, with”
Lines 161 – 162: “with 161 metamorphic and sedimentary rocks from five geologic period including
the Archean, 162 Proterozoic, Paleozoic, Mesozoic and Cenozoic epochs.”
Please reformulate the phrase.
Line 181: “better than 1 m” What the authors refer to? Can the author specify the exact height
precision?
Line 184: “no big changes. so that Consequently, the reference DSM is was considered sufficient to
validate for validation purposes of our radargrammetric method.”
Line 190: “matching, as specified in Section 2.2.”
Line 206: “It can be seen that most of the”
Line 214: “which are shown in show as red or blue color.”
Line 215: “Thus, stereo matching becomes would be easier and faster.”
Line 296: “with a step size of 20, and the”
Line 299: “completeness of the disparity map,”
Line 300: “disparity values, while the effective”
Line 359: “resampled to a 10 m spatial resolution”
Line 361: “airborne DSM is based on WGS84 ellipsoid”
Lines 557 – 564: I recommend inserting a space after values: for example: 36.6-38.6m - 36.6-38.6 m
and keeping the same structure in the whole document.
Line 642: “to the SRTM DEM owing due to the”
Line 648: “will be fused combined to improve the height”

Author Response

Thank you very much for your meticulous reading and pointing out the grammer errors. The questions and suggestions has been listed with my answers in the following:

  • From the methodological point of view the paper is presented in a very attractive manner. My general comment is that the authors should consider improving the language and phrasing.

Thank you very much. We have improved the language and phrasing as you can see from the changes marked up using the “Track Changes” function in the updated version of the manuscript.

  • Lines 13 – 15: “Stereo image matching in radargrammetry is the process of deter-13 mining correspondences and the performance of image matching influences the final 14 quality of DSM used for spatial-temporal analysis of landscapes and terrain”. Please reformulate the phrase.

The sentence is revised to be “Stereo image matching in radargrammetry refers to the process of determining homologous points in two images. The performance of image matching influences the final quality of DSM used for spatial-temporal analysis of landscapes and terrain.” as you can see in line 13-16 of the updated version in the manuscript.

  • Line 16: “methods are most commonly-used but and usually produce”

The sentence is revised to be “methods are commonly-used but usually produce” as you can see in line 16-17 of the updated version in the manuscript.

  • Line 19: “To fill this gap, we apply applied a hierarchical”

The sentence is revised to be “To fill this gap, we propose a hierarchical” as you can see in line 19-20 of the updated version in the manuscript.

  • Line 35: “Since the early 2000s, global SAR-based terrain models have been acquired from SAR images imagery.”

The sentence is revised to be “Synthetic Aperture Radar (SAR) images can be used to construct digital surface models.” as you can see in line 38-39 of the updated version in the manuscript.

  • Line 36: “For instance, SRTM and TanDEM-X DEM are the most representative global DEMs.”

The sentence is revised to be “For instance, SRTM DEM [2] and TanDEM-X DEM [3] are the most representative InSAR products, while StereoSAR DSMs are generated from SAR images with different incidence angles.” as you can see in line 44-46 of the updated version in the manuscript.

  • Line 46: “given pair of images usually either InSAR or radargrammetry technique can…”

The sentence is revised to be “For a given pair of images usually either an InSAR or radargrammetry technique can be applied: the former requires small changes in illumination angle to acquire repeat-pass images, whereas the latter requires large baselines to yield stable intersection of line-of-sight rays.” as you can see in line 51-55 of the updated version in the manuscript.

  • Line 54: “local matching algorithms and global matching algorithms.”

The sentence is revised to be “local matching and global matching algorithms” as you can see in line 60 of the updated version in the manuscript.

  • Line 69: “The dependence of windows implicitly assumes that disparities of all”

The sentence is revised to be “The use of a support windows implicitly assumes that the disparities between the pixels within the window are consistent.” as you can see in line 75-76 of the updated version in the manuscript.

  • Line 113: “DATE workflow [26-28], which exploits the Semi-Global Block Matching (SGBM) algorithm”

The sentence is revised to be “In 2016, Di Rita et. al. developed a radargrammetric DSM workflow termed the DATE workflow [26-28] that exploits the OpenCV library Semi-Global Block Matching (SGBM) algorithm [29,30].” as you can see in line 119-121 of the updated version in the manuscript.

  • Lines 116–117: “Whereas the workflow does not transfer disparity from the higher level to a lower level to reduce the disparity search range for a lower-level matching, instead it converts the disparities to height correction values at every level” Please reformulate the phrase.

The sentence is revised to be “The DATE workflow does not transfer disparity from the higher level to a lower level to reduce the disparity search range for a lower-level matching; instead, it converts the disparities to height correction values at every level.” as you can see in line 13-16 of the updated version in the manuscript.

  • Line 127: “where only two penalty parameter values need are needed/are necessary to be set.” Please reformulate the phrase.

The sentence is revised to be “The pipeline was built in a user-friendly manner where only two penalty parameter values need to be set.” as you can see in line 141-142 of the updated version in the manuscript.

  • Line 133: “approach were also included to compare in the vertical precision comparation”

The sentence is revised to be “NASA’s 30 m resolution SRTM DEM and DSMs extracted by hierarchical NCC matching approach were also included to compare the vertical accuracy.” as you can see in line 146-148 of the updated version in the manuscript.

  • Line 134: “not only produces produce DSMs with higher”

The sentence is revised to be “not only produces DSMs with higher vertical accuracy” as you can see in line 148-149 of the updated version in the manuscript.

  • Lines 137–138: “edge detector is more suitable than the constant penalty and the gray gradient penalty 137 functions used in stereo SAR image matching.” Please reformulate the phrase.

The sentence is revised to be “In addition, a penalty function exploiting the Canny edge detector delivers higher vertical accuracy than the constant penalty function, or the gray gradient penalty function in stereo SAR semi-global matching.” as you can see in line 151-154 of the updated version in the manuscript.

  • Lines 148–149: “based on the Range-Doppler model to derive the 3D geographic coordinates, and a triangulation interpolation yields a final gridded DSM.” Please reformulate the phrase.

The sentence is revised to be “In turn, these homologous points are intersected based on the Range-Doppler model to derive 3D geographic coordinates, so the coordinates are used to yield a final gridded DSM by a triangulation interpolation.” as you can see in line 166-168 of the updated version in the manuscript.

  • Line 150: “function of SGM as will be shown in Equation (1),”

The sentence is revised to be “The formulation of the penalty function as expressed in Equation (1) influences the matching performance (discussed in section 2.4).” as you can see in line 168-170 of the updated version in the manuscript.

  • Line 160: “from 150 m to 1512 m above sea level, Mount Song possesses is characterized by a complex”

The sentence is revised to be “from 150 m to 1512 m above sea level, Mount Song possesses a complex” as you can see in line 180 of the updated version in the manuscript.

  • Line 161-162: “geological history of over 3.6 billion years, with 161 metamorphic and sedimentary rocks from five geologic period including the Archean, 162 Proterozoic, Paleozoic, Mesozoic and Cenozoic epochs.” Please reformulate the phrase.

The sentence is revised to be “Mount Song, with 3.6 billion years of geological history, is composed of metamorphic and sedimentary rocks from five geologic periods including the Archean, Proterozoic, Paleozoic, Mesozoic, and Cenozoic epochs.” as you can see in line 181-183 of the updated version in the manuscript.

  • Line 181: “better than 1 m” What the authors refer to? Can the author specify the exact height precision?

Yes, the height precision of the airborne reference DSM is 1m. The sentence is revised to be “An airborne photogrammetric DSM at 1 m resolution with a height precision of 1 m was used as the reference data for evaluating the quality of our radargrammetric DSMs.” as you can see in line 199-201 of the updated version in the manuscript.

  • Line 184: “no big changes. so that Consequently, the reference DSM is was considered sufficient to validate for validation purposes of our radargrammetric method.”

The sentence is revised to be “no big changes so that the reference DSM is sufficient to validate our radargrammetric method” as you can see in line 204-205 of the updated version in the manuscript.

  • Line 190: “matching, as specified in Section 2.2.”

The sentence is revised to be “matching, as discussed in the next subsection.” as you can see in line 214 of the updated version in the manuscript.

  • Line 206: “It can be seen that most of the”

The sentence is revised to be “Only one color showing in the overlapping areas, indicates that there is no disparity between the homologous points, and most of the disparities were eliminated in each epipolar pair.” as you can see in line 242-244 of the updated version in the manuscript.

  • Line 214: “which are shown in show as red or blue color.”

The sentence is revised to be “Only slight residual disparities remain along the epipolar direction shown in the red or blue color.” as you can see in line 251-252 of the updated version in the manuscript.

  • Line 215: “Thus, stereo matching becomes would be easier and faster.”

The sentence is revised to be “Thus, stereo matching is easier and faster.” as you can see in line 253 of the updated version in the manuscript.

  • Line 296: “with a step size of 20, and the”

Thank you, we lost the “of” in the old version. Now the sentence is revised to be “with a step size of 20, and the” as you can see in line 339 of the updated version in the manuscript.

  • Line 299: “completeness of the disparity map,”

Thank you, we lost a “the” in the old version. The sentence is revised to be “the completeness of the disparity map” as you can see in line 342 of the updated version in the manuscript.

  • Line 300: “disparity values, while the effective”

The sentence is revised to be “disparity values. While the effective” as you can see in line 343 of the updated version in the manuscript.

  • Line 359: “resampled to a 10 m spatial resolution”

Thank you, we lost the “a” in the old version. The sentence is revised to be “resampled to a 10 m spatial resolution” as you can see in line 404 of the updated version in the manuscript.

  • Line 361: “airborne DSM is based on WGS84 ellipsoid”

The sentence is revised to be “The height datum of radargrammetric DSMs and the reference airborne DSM is WGS84 ellipsoid” as you can see in line 406-407 of the updated version in the manuscript.

  • Lines 557 – 564: I recommend inserting a space after values: for example: 36.6-38.6m - 36.6-38.6 m and keeping the same structure in the whole document.

Thank you, the sentence is revised to be “Specifically, the hierarchical NCC method yielded RMSE values of 22.5-24.4 m and LE90 values of 37.2-40.2 m in mountainous areas, with RMSE values of 20.8-21 m and LE90 values of 36.6-38.6 m in flat areas. The hierarchical SGM method provides RMSE values of 8.9-11.6 m and LE90 values of 12.5-17.9 m in mountainous areas, with RMSE values of 2.0-2.3 m and LE90 values of 3.6-4.1 m in flat areas.” as you can see in line 613-617.

 We also have changed inserted a space after values as suggested throughout the paper, as you can see in the updated version of the manuscript.

  • Line 642: “to the SRTM DEM owing due to the”

The sentence is revised to be “to the SRTM DEM owing to the” as you can see in line 702 of the updated version in the manuscript.

  • Line 648: “will be fused combined to improve the height”

The sentence is revised to be “will be fused to improve” as you can see in line 708 of the updated version in the manuscript.

Reviewer 3 Report

The manuscript is titled Radargrammetric DSM Generation by Semi-Global Matching and Evaluation of Penalty Functions seems interesting and well written after minor revision can be accepted for publication. Some comments and questions are as follow;

  • What is the main input of this manuscript please clearly present
  • Did you compare between Asc and Des datasets? or just used one pair?
  • Why use the Lee filter for speckle noise?
  • Some parts look like a shadow how did you solve it?
  • The conclusion does not well support the result
  • English error and grammatical checks may be applied 

Author Response

Thank you very much for the suggestive comments. The questions and suggestions has been listed with my answers in the following:

  • What is the main input of this manuscript please clearly present?

The manuscript is aimed at investigating the feasibility and effectiveness of semi-global dense matching (SGM) algorithm to generate high-quality radargrammetric DSMs. In addition, as a module of SGM algorithm, we investigate which penalty function performs better in stereo SAR semi-global matching. We provide a powerful alternative tool to local matching algorithms with higher accuracy and efficiency.

As suggested, the abstract is revised as you can see in line 11-33 of the updated version in the manuscript: “Radargrammetry is a useful approach to generate Digital Surface Models (DSMs) and an alternative to InSAR techniques that are subject to temporal or atmospheric decorrelation. Stereo image matching in radargrammetry refers to the process of determining homologous points in two images. The performance of image matching influences the final quality of DSM used for spatial-temporal analysis of landscapes and terrain. In SAR image matching, local matching methods are commonly-used but usually produce sparse and inaccurate homologous points adding ambiguity to final products; global or semi-global matching methods are seldom applied even though more accurate and dense homologous points can be yielded. To fill this gap, we propose a hierarchical semi-global matching (SGM) pipeline to reconstruct DSMs in forested and mountainous regions using stereo TerraSAR-X images. In addition, three penalty functions were implemented in the pipeline and evaluated for effectiveness. To make accuracy and efficiency comparisons between our SGM dense matching method and the local matching method, the normalized cross-correlation (NCC) local matching method was also applied to generate DSMs using the same test data. The accuracy of radargrammetric DSMs was validated against an airborne photogrammetric reference DSM and compared with the accuracy of NASA’s 30m SRTM DEM. The results show the SGM pipeline produces DSMs with height accuracy and computing efficiency that exceeds the SRTM DEM and NCC-derived DSMs. The penalty function adopting the Canny edge detector yields a higher vertical precision than the other two evaluated penalty functions. SGM is a powerful and efficient tool to produce high-quality DSMs using stereo Spaceborne SAR images.”

And the introduction is revised as you can see in line 136-156 of the updated version in the manuscript: “In this study, we propose a hierarchical SGM dense matching pipeline to generate high quality radargrammetric DSMs in densely vegetated and mountainous areas. The feasibility and effectiveness of the semi-global dense matching algorithm is investigated using Stripmap and Spotlight mode TerraSAR-X stereo data pairs covering Mount Song in central China. Furthermore, we investigated the influence of three penalty functions on the vertical accuracy of final DSMs. The pipeline was built in a user-friendly manner where only two penalty parameter values need to be set. Without any post-processing, the disparity map can be directly used for generating final DSMs. A high-resolution airborne photogrammetric DSM was used to validate the radargrammetric DSMs. NASA’s 30 m resolution SRTM DEM and DSMs extracted by hierarchical NCC matching approach were also included to compare the vertical accuracy. The results demonstrate that, our SGM pipeline not only produces DSMs with higher vertical accuracy than the SRTM DEM and NCC-derived DSMs, but is also more efficient than the NCC local matching method. In addition, a penalty function exploiting the Canny edge detector delivers higher vertical accuracy than the constant penalty function, or the gray gradient penalty function in stereo SAR semi-global matching. Semi-global matching is a powerful alternative tool to local matching algorithms with higher accuracy and efficiency for radargrammetric DSM generation in complex mountainous areas.”

  • Did you compare between Asc and Des datasets? or just used one pair?

Yes, we used two stereo pairs, one is the Spotlight mode image pair, which is collected from a descending orbit, and the other is the Stripmap mode image pair, which is collected from an ascending orbit. Test area 1 is only covered with the Stripmap pair, i.e., the descending data. While test area 2 and test area 3 are all covered by both pairs, we have showed and compared the height accuracy values and hillshades for test area 2 in section 3.3.2 and for test area 3 in section 3.3.3.

  • Why use the Lee filter for speckle noise?

Any commonly-used filter for SAR images can be used to reduce the noise and speckle. We used Lee filter for our experiments, but other filters such as Refined Lee filter or Frost filter can also be applied. The choice of different filters won’t make a huge difference to the result.

  • Some parts look like a shadow how did you solve it?

At this stage, shadow areas on the epipolar images are lacking in texture information just like the layover case. Only sparse homologous points can be found in such area as shown in Figure 9. This is the problem we will further investigate and find a proper way to solve in the future.

  • The conclusion does not well support the result.

As suggested, the conclusion is revised in line 687-709 in the updated version in the manuscript:

“In this paper, we implemented a hierarchical SGM dense matching pipeline for stereo SAR image matching, which can produce high quality radargrammetric DSMs in forested mountain areas. The pipeline is designed in a user-friendly manner with only two user-set penalty parameter values. The output disparity maps can be directly used to generate high-quality DSMs without post-processing. The SGM-derived DSMs, NCC-derived DSMs, and the 30 m resolution SRTM DEM were analyzed in terms of accuracy and efficiency. The results show that in mountainous regions, the height accuracy of the SGM-derived DSM is higher than that of the SRTM DEM; in flat regions, the height accuracy of the SGM-derived DSM is at the same level as the SRTM DEM.  The NCC matching method yielded height accuracy twice that of the SRTM DEM. Our SGM dense matching pipeline provides the highest vertical accuracy and processing efficiency. For severe layover areas, the number of successfully matched homologous points is inadequate as the texture information is insufficient; but the height accuracy of the SGM-derived DSMs is still close to the SRTM DEM owing to the regularization constraints. Furthermore, three penalty functions were evaluated in our SGM matching pipeline. The penalty function of the canny edge detector yields the highest vertical accuracy. In the future, efforts will be devoted to the development of new penalty function more suitable for SAR image matching. Additionally, different matching costs such as census, NCC and mutual information will be evaluated. Stereo DSMs generated from different orbits will be fused to improve the height accuracy in layover areas.”